



# An EnKF-based ocean data assimilation system improved by adaptive observation error inflation (AOEI)

Shun Ohishi[1, 2, 3], Takemasa Miyoshi[1, 2, 4], and Misako Kachi[5]

[1]RIKEN Center for Computational Science, Kobe, 6500047, Japan
[2]RIKEN Cluster for Pioneering Research, Kobe, 6500047, Japan
[3]Institute for Space-Earth Environmental Research, Nagoya University, Nagoya, 4648601, Japan
[4]RIKEN Interdisciplinary Theoretical and Mathematical Sciences Program (iTHEMS), Kobe, 6500047, Japan
[5]Earth Observation Research Center, Japan Aerospace Exploration Agency, Tsukuba, 3058505, Japan

*Correspondence to*: Shun Ohishi (shun.ohishi@riken.jp) and Takemasa Miyoshi (takemasa.miyoshi@riken.jp)

**Abstract.**

A previous study proposed an adaptive observation error inflation (AOEI) method for an ensemble Kalman filter-based atmospheric data assimilation system to assimilate all-sky infrared brightness temperatures. Brightness temperature differences between clear- and cloudy-sky radiances are large, and observation-minus-forecast differences or innovations are therefore
likely to be large around boundaries between clear- and cloudy-sky regions. The AOEI method mitigates these discrepancies by adaptively inflating observation errors. Ocean frontal regions have similar characteristics to the borders between clear- and cloudy-sky regions with large innovations. Consequently, we have implemented the AOEI with an EnKF-based regional ocean data assimilation system, in which the assimilation interval is set to 1 day to utilize frequent satellite observations. We conducted sensitivity experiments to investigate the impacts of the AOEI on salinity structure, geostrophic balance, and
accuracy. A control run, in which the AOEI is not applied, shows the degradation of low-salinity North Pacific Intermediate Water around the Kuroshio Extension region, where the innovation amplitude and forecast ensemble spread are large in association with the fronts and eddies. The resulting large temperature and salinity increments weaken the density stratification, leading to large vertical diffusivity. As a result, the low salinity water in the intermediate layer is lost through strong vertical diffusion. When the AOEI is used, the salinity structure in the ocean interior is preserved because the AOEI suppresses the
salinity degradation by reducing the temperature and salinity increments. We also demonstrate that the AOEI provides significant improvement of the geostrophic balance and the accuracy of temperature, salinity, and surface flow fields.

**Short summary (487/500 character)**

An adaptive observation error inflation (AOEI) method was proposed for atmospheric data assimilation to mitigate erroneous
analysis updates caused by large observation-minus-forecast differences for satellite brightness temperature around clear- and cloudy-sky boundaries. This study implemented the AOEI with an ocean data assimilation system, leading to an improvement of analysis accuracy and dynamical balance around the frontal regions with large meridional temperature differences.





## 1. Introduction

The ensemble Kalman filter (EnKF) estimates flow-dependent forecast errors from an ensemble of model forecasts, and
calculates the best estimates (i.e., analyses) bv combining forecasts and observations with their error covariances (Evensen,
1994, 2003). The EnKF has the advantage of being easy to implement for various models, but it has been used in only two
ocean reanalysis datasets thus far (Balmaseda et al., 2015; Martin et al., 2015): the Predictive Ocean Atmosphere Model for
Australia (PAOMA) Ensemble Ocean Data Assimilation System (PEODAS; Yin et al., 2011) and TOPAZ4 (Sakov et al.,
2012). In contrast, the three-dimensional variational method (3D-VAR) is the most widely used in ocean data assimilation
systems.

With the enhancement of in-situ and satellite observations, the number of observations has increased dramatically.
Argo profiling float observations since the 2000s provides a large number of in-situ temperature and salinity data in the ocean
interior. Although satellite sea surface salinity (SSS) data since 2010 are relatively inaccurate, particularly in coastal and high-
latitude regions (Abe and Ebuchi, 2014), previous studies have demonstrated the positive effects of SSS assimilation on the
analyses of ocean interior structures such as mixed and barrier layers (Chakraborty et al., 2015), low salinity water caused by
river discharge (Toyoda et al., 2015), and El Niño-Southern Oscillation prediction (Hackert et al., 2011). A Japanese
geostationary satellite, Himawari-8 (Bessho et al., 2016; Kurihara et al., 2016), has observed sea surface temperatures (SSTs)
in the Pacific region at a high spatiotemporal resolution of 2 km and 10 minutes since July 2015. The Surface Water and Ocean
Topography (SWOT) satellite is scheduled to be launched in 2022 and will provide high-resolution and two-dimensional sea
surface height (SSH) anomalies.

For effective use of dense and frequent satellite observations, Ohishi et al. (in review) performed sensitivity
experiments using an EnKF-based ocean data assimilation system with an assimilation interval of 1 day, which is more frequent
than the five- and seven-day intervals in the existing EnKF-based systems (PEODAS and TOPAZ4, respectively). They
demonstrated that the combination of incremental analysis updates (IAU; Bloom et al., 1996) and relaxation-to-prior
perturbations (RTPP; Kotsuki et al., 2017; Zhang et al., 2004) to restore the forecast ensemble perturbations toward the analysis
by 80%–90% produced optimal results in terms of both dynamic balance and accuracy. However, their system contained
several tuning parameters such as observation errors, ensemble size, and localization scale. Previous studies have prescribed
observation errors in various ways, e.g., using spatiotemporally fixed constants (Miyazawa et al., 2012; Xu and Oey, 2014);
assuming them to be standard deviations calculated from historical observations (Miyazawa et al., 2009; Usui et al., 2006);
estimating them from other assimilation datasets (Penny et al., 2013); and assuming that observation error covariance matrices
are proportional to the forecast error covariance matrices (Carton et al., 2018; Yin et al., 2011). A technique to inflate the
observation errors based on the innovation statistics (Desroziers et al., 2006) is called an adaptive observation error inflation
(AOEI) method and was recently proposed for assimilating all-sky infrared satellite brightness temperatures in an atmospheric
data assimilation system (Minamide and Zhang, 2017; Zhang et al., 2016). Since the brightness temperature differences
between clear- and cloudy-sky radiances are large, there are large observation-minus-forecast differences or innovations
around boundaries between clear- and cloudy-sky regions. This results in erroneous analysis increments and degrades the





analysis. The AOEI mitigates the large discrepancies between forecasts and observations by adaptively inflating the observation errors. As shown in Section 3, an EnKF-based ocean data assimilation system used in this study shows that the salinity structure in the ocean interior is degraded around the Kuroshio Extension (KE) region, where the spatiotemporal

variations are large due to fronts and eddies. Ocean frontal regions have similar characteristics to the borders between clear- and cloudy-sky regions with large innovations, and the AOEI method is therefore expected to be useful for improving EnKF-based ocean data assimilation systems.

This study aims to investigate the causes of the salinity degradation around the KE region, and to evaluate the impacts of the AOEI on the salinity structure, dynamical balance, and accuracy. The remainder of this paper is organized as follows:

details of the AOEI method, the experiment design, and the methods to evaluate geostrophic balance and accuracy in sensitivity experiments are presented in section 2; section 3 describes the causes of the salinity degradation in the intermediate layer and the positive impacts of the AOEI on the geostrophic balance and accuracy for temperature, salinity, and surface flow. A summary is provided in section 4.

## 2. Methods

### 2.1. Adaptive observation error inflation (AOEI)

Manual tuning of observation errors is computationally expensive, and several studies have proposed adaptive estimation methods using the innovation statistics of Desroziers et al. (2006):

$$\boldsymbol{d_b^o}(\boldsymbol{d_b^o})^\mathrm{T} \approx \mathbf{H}\mathbf{P^b}\mathbf{H^T} + \mathbf{R}. \tag{1}$$

Here, $\boldsymbol{d_b^o}$ ($= \boldsymbol{y} - \mathbf{H}\overline{\boldsymbol{x^b}}$) is an innovation vector, where $\boldsymbol{y}$, $\mathbf{H}$, and $\overline{\boldsymbol{x^b}}$ denote an observation vector, linear observation operator, and forecast ensemble mean state vector, respectively. $\mathbf{P^b}$ and $\mathbf{R}$ are the forecast and observation error covariance matrices, respectively. Expressing Eq. (1) in a scalar form, the observation error $\sigma_{est-o}$ may be estimated by


$$\sigma_{est-o}^2 = (d_b^o)^2 - \sigma_{H(x^b)}^2, \tag{2}$$

where $\sigma_{H(x^b)}$ is the forecast ensemble spread in observation space. In the AOEI method (Minamide and Zhang, 2017; Zhang et al., 2016), larger observation errors $\sigma_o$ between the estimated and prescribed errors are used:


$$\sigma_o^2 = \max\{\sigma_{pre-o}^2, \sigma_{est-o}^2\}, \tag{3}$$



where $\sigma_{pre-o}$ is the prescribed observation error. As described in section 1, the AOEI suppresses erroneous analysis increments associated with systematic errors, biases, and representation errors by adaptively inflating the observation errors when the
squared innovation is larger than the sum of the prescribed observation and ensemble-based forecast error variances.

## 2.2. Experimental design

This study uses an EnKF-based regional ocean data assimilation system known as sbPOM-LETKF (Ohishi et al. in review), comprising a sigma-coordinate regional ocean model, the Stony Brook Parallel Ocean Model version 1.0 (sbPOM; Jordi and
Wang, 2012; Ohishi et al. in review), and a three-dimensional local ensemble transform Kalman filter (3D-LETKF; Hunt et al., 2007; Miyoshi and Yamane, 2007). The sbPOM is configured for the northwestern Pacific region (117°–180°E, 15°–50°N) with horizontal resolution of 0.25° and 50 sigma layers. The bottom topography is taken from a 1 arc-minute global relief model of Earth's surface (ETOPO1; Amante and Eakins, 2009) and is smoothed by a Gaussian filter with a 200 km e-folding scale to reduce pressure gradient errors at steep bottom slopes (Mellor et al., 1994). Monthly (seasonal) temperature and salinity
climatologies from the World Ocean Atlas 2018 (WOA18; Locarnini et al., 2019; Zweng et al., 2019) with horizontal resolution of 1° and 57 (103) layers are used for the initial conditions over depths shallower (deeper) than 1500 m. Lateral boundary conditions for temperature, salinity, and horizontal velocity are derived from the Simpler Ocean Data Assimilation (SODA) version 3.7.2 (Carton et al., 2018) with horizontal resolution of 0.5° and 50 layers. The Japanese 55-year Reanalysis (JRA55; Kobayashi et al., 2015) with horizontal and temporal resolution of 1.25° and 6 hours, respectively, is adopted for the
atmospheric boundary conditions including air temperature and specific humidity at 2 m, wind velocity at 10 m, shortwave radiation, total cloud fraction, sea level pressure, and precipitation. River discharge is obtained from the Japan Aerospace Exploration Agency (JAXA)'s land surface and river simulation system, Today's Earth (TE)-Global (https://www.eorc.jaxa.jp/water/), with horizontal and temporal resolution of 0.25° and 3 hours, respectively. To avoid filter divergence, the atmospheric and lateral boundary conditions other than rainfall and river discharge are perturbed in the same
way as in Ohishi et al. (in review). The model with 100 ensemble members is spun up from 1 January 2011 to 6 July 2015, using the initial conditions with no motion. During the spin-up period, simulated temperature and salinity are nudged towards the monthly climatology from the WOA18 with a 90-day timescale to prevent northward overshoot of the Kuroshio along the east coast of Japan.

       The LETKF with 100 ensemble members is used to assimilate the following observations on a 1-day assimilation
interval: satellite SSTs from Himawari-8 (Bessho et al., 2016; Kurihara et al., 2016) and the Global Change Observation Mission–Water (GCOM-W: https://gportal.jaxa.jp/gpr/?lang=en); satellite SSS from Soil Moisture and Ocean Salinity (SMOS: http://www.esa.int/Applications/Observing_the_Earth/SMOS) and Soil Moisture Active Passive (SMAP) version 4.3 (Meissner et al., 2018); SSH estimated by summing satellite SSH anomalies from the Copernicus Marine Environment Monitoring Service (CMEMS: https://marine.copernicus.eu/) and mean dynamic ocean topography obtained by averaging the
simulated SSH over 2012–14; and in-situ temperatures and salinity from the Global Temperature and Salinity Profile Programme (GTSPP; Sun et al., 2010) and Advance automatic QC (AQC) Argo Data version 1.2a (AQC:





http://www.jamstec.go.jp/ARGO/argo_web/argo/?page_id=100&lang=en). Covariance localization is applied using the Gaussian function with horizontal and vertical localization scales $LS = 300\ km$ and $100\ m$, respectively, following Miyazawa et al. (2012) and Penny et al. (2013). We assume that the localization function becomes zero beyond $2\sqrt{10/3}\ LS \approx$

1100 km (370 m) in the horizontal (vertical) direction (Miyoshi et al., 2007). The prescribed observation errors for temperature, salinity, and SSH are set to 1.0 °C, 0.3, and 0.2 m, respectively. We adopt the combination of the IAU (Bloom et al., 1996) and RTPP (Kotsuki et al., 2017; Zhang et al., 2004), in which the analysis ensemble perturbations are relaxed toward the forecast ensemble perturbations by 90 % while maintaining the analysis ensemble mean, because the sensitivity experiments in Ohishi et al. (in review) demonstrated that it results in the best dynamical balance and accuracy. Although this

may not be optimal, our computational resources are limited, and thus the RTPP relaxation parameter is fixed at 90%.

To evaluate the impacts of the AOEI on the ocean salinity structure, dynamical balance, and accuracy, we conduct AOEI and control (CTL) runs with and without applying the AOEI, respectively, from the start date of the Himawari-8 observation (7 July 2015) to 31 December 2015. During the assimilation period, the SSS nudging with a 90-day timescale is applied to prevent a surface freshening drift.


## 2.3 Evaluation method

As in Ohishi et al. (in preparation), this study evaluates geostrophic balance and accuracy using the nonlinear balance equation (NBE) and root mean square deviations (RMSDs) relative to observations, respectively (see subsections 2.3.1 and 2.3.2).

### 2.3.1 Nonlinear balance equation

For the analysis fields, the geostrophic balance equation is represented as

$$f\boldsymbol{k} \times \delta\boldsymbol{u} = -g\boldsymbol{\nabla}_{\boldsymbol{h}}\delta\eta, \tag{4}$$

where $f$ is the vertical component of the Coriolis parameter, $\boldsymbol{k}$ is a unit vector in the vertical direction, $\delta$ is the analysis increment, $\boldsymbol{u} = (u, v)$ denotes horizontal velocity at the sea surface, $g = 9.8\ m\ s^{-2}$ is gravitational acceleration, $\boldsymbol{\nabla}_{\boldsymbol{h}} = (\partial/\partial x, \partial/\partial y)$ is the horizontal gradient operator, and $\eta$ denotes SSH. By taking $\partial/\partial x$ of the x-component and $\partial/\partial y$ of the y-component of Eq. (4), the geostrophic equation can be reduced to the nonlinear balance equation (NBE; Shibuya et al., 2015; Zhang et al., 2001):


$$-f\delta\zeta + \beta\delta u + g\boldsymbol{\nabla}_{\boldsymbol{h}}^{2}\delta\eta = 0, \tag{5}$$

where $\zeta = \partial v/\partial x - \partial u/\partial y$ is the relative vorticity at the sea surface and $\beta = \partial f/\partial y$ is the planetary vorticity gradient. If the analysis fields do not satisfy geostrophic balance, there is an absolute NBE residual:






$$\Delta NBE \equiv abs\left(-f\delta\zeta + \beta\delta u + g\boldsymbol{\nabla}_h^2\delta\eta\right), \tag{6}$$

where $abs$ indicates taking the absolute value. A smaller (larger) $\Delta NBE$ indicates more (less) geostrophic balance.

**2.3.2 RMSD**


We evaluate the accuracy using the RMSDs calculated relative to the following observations: in-situ temperature and salinity over 1–525 m depth and in-situ horizontal velocity over 8–36 m depth at (144.6°E, 32.3°N) south of the KE from the Kuroshio Extension Observatory (KEO) buoy (https://www.pmel.noaa.gov/ocs/; See Fig. 15), SSH and SSH anomaly gridded datasets with horizontal resolution of 0.25° from Archiving, Validation and Interpretation of Satellite Oceanographic data (AVISO;

Ducet et al., 2000), and in-situ surface horizontal velocity from surface drifter buoys of the Global Drifter Program (Elipot et al., 2016). We note that the AVISO is not an independent dataset because satellite SSH anomalies are used in this system, whereas the KEO and surface drifter buoys are independent observations. The validation in the ocean interior in this study is limited due to the paucity of available independent observations.

In this study, we calculate the $\Delta NBE$ and RMSDs using daily outputs from the CTL and AOEI runs. To compare the

AOEI run with the CTL run, we also calculate improvement ratios *IR* for $\Delta NBE$ and RMSD:

$$IR_{NBE} = \frac{(\Delta NBE)_{CTL} - (\Delta NBE)_{AOEI}}{(\Delta NBE)_{CTL}} \times 100 \text{ and} \tag{7}$$

$$IR_{RMSD} = \frac{(RMSD)_{CTL} - (RMSD)_{AOEI}}{(RMSD)_{CTL}} \times 100, \tag{8}$$

where the subscripts $CTL$ and $AOEI$ indicate the CTL and AOEI runs, respectively. Using the bootstrap method with 10,000 cycles, we detect significant improvement and degradation in the AOEI run relative to the CTL run at the 99% confidence level.

**3. Results**

**3.1. Salinity degradation in the NPIW around the KE region**

As shown in Fig. 1, the SST field in the CTL run agrees well with the satellite observations. Although the satellite-derived SSS has large errors, especially in coastal and high-latitude regions (Abe and Ebuchi, 2014), the SSS spatial pattern appears to be reproduced well in the analysis field (Fig. 2). However, the CTL run has noisier signals in the latter half of the experiment period, particularly in the SSS analysis fields. We also assess the monthly-mean temperature $T$, salinity $S$, and

potential density $\sigma_\theta$ along 150°E and 35°N sections across the KE (Figs. 3, 4). During the initial stages of the experiment period, the North Pacific Intermediate Water (NPIW), characterized by low minimum salinity, is distributed within $\sigma_\theta = 26.5$–27.25 kg m$^{-3}$ (Talley, 1993; Yasuda, 1997). However, as the assimilation period progresses, the low salinity structure in the





intermediate layer around the KE region is lost along with the noisy signals, whereas the temperature structure persists with lower temperatures at deeper depths.

To investigate the cause of the salinity degradation, we calculate the salinity budget equation in the intermediate layer around the KE region (140°–160°E, 30°–40°N, 500–1000 m depth):

$$\frac{\partial S}{\partial t} = \nabla(\boldsymbol{\kappa} \circ \nabla S) - \boldsymbol{v} \cdot \nabla S + (increment), \tag{9}$$

where $\nabla = (\partial/\partial x, \partial/\partial y, \partial/\partial z)$ denotes the three-dimensional gradient operator, $\boldsymbol{\kappa} = (\kappa_x, \kappa_y, \kappa_z)$ is a diffusivity vector, ∘ indicates a Schur product, $\boldsymbol{v} = (u, v, w)$ is three-dimensional velocity, and $(increment)$ indicates the analysis increments. Equation (9) does not include a residual term because the system conserves each term of the salinity budget equation. Figure 5a indicates that the salinity tendency term [the left-hand side (LHS) term of Eq. (9)] is positive and corresponds to the salinity increase shown in Figs. 3 and 4. The positive salinity tendency term is caused mainly by the diffusion term [the first term on

the right-hand side (RHS) of Eq. (9)] and partly by the advection term [the second term on the RHS of Eq. (9)]. The diffusion term is dominated only by the vertical diffusion, and the horizontal diffusion makes almost no contribution (Fig. 5b). The advection term consists of different components in different months during the experiment period (Fig. 5c): meridional advection in July, zonal and meridional advection in August, and zonal and vertical advection in September–December 2015. In contrast, the analysis increment term [the last term on the RHS of Eq. (9)] has only a minor impact but plays a role in

restoring the low salinity water. Therefore, the vertical diffusion is the main cause of the salinity degradation in the intermediate layer around the KE region.

**3.2. Spatiotemporal characteristics of the AOEI in the surface fields**

To investigate how much the AOEI applies to the SST, SSS, and SSH fields, we calculate the monthly-mean ratio of the area

where the AOEI is applied to the entire system domain (Fig. 6). Application of the AOEI to the SSS field is the highest around 35%–40% of the domain because the instantaneous satellite observations are noisy (figure not shown). The AOEI is also applied to the SST field at a relatively high ratio of 5%–10%, whereas the ratio in the SSH field is exceedingly small (less than 0.1%). This indicates that the AOEI method is applied substantially to the SST and SSS fields, and rarely to the SSH field.

    We also examine the spatial characteristics of where the AOEI is applied to the SST and SSS fields by calculating

the ratio of the period when the AOEI method is applied compared with the total experiment period (Figs. 7, 8). High SST ratios are distributed in the coastal and frontal regions, including the Kuroshio, the KE, and a subpolar front along J1 around 150°E, 40°N (Isoguchi et al., 2006; Kida et al., 2015; Fig.7a). The SSS ratios are high in the East China Sea, Japan Sea, and high-latitude regions (Fig. 8a). The spatial pattern of the positive and negative innovation phases is asymmetric in both the SST and SSS fields (Figs. 7b, c, 8b, c). In the positive innovation phase, the high SST ratios are distributed only along the

northeastern coast of Japan at 140°–150°E, 40°–50°N (Fig. 7b), whereas in the negative innovation phase, high SST ratios are





more widely distributed, covering coastal and frontal regions (Fig. 7c). In the negative innovation phase, the SSS ratios are higher in the East China Sea, Japan Sea, and high-latitude regions (Fig. 8b, 8c). In the SST and SSS fields, the spatial patterns of the positive forecast biases correspond closely to the high ratios in the negative innovation phase (Figs. 7c, 8c, 9). Therefore, the forecast SST and SSS biases lead to the asymmetry in which the AOEI is applied more during negative innovation phases

than during positive phases, as seen in Fig. 6a, 6b.

In the SST field, large innovation amplitude and forecast ensemble spread are distributed along the KE and the J1. In the SSS field, the ensemble spread is large along the KE and J1, where the salinity innovation amplitude is large and exceeds 1.0. This demonstrates that large temperature and salinity analysis increments are likely to be generated in the KE and J1 regions if the AOEI is not applied, as in the CTL run.


### 3.3. Improvements of the salinity structure by the AOEI

We compare monthly temperature and salinity fields between the CTL and AOEI runs at the sea surface and along the 150°E and 35°N sections. In the AOEI run, noisy signals are reduced in the temperature and salinity fields at the sea surface (figure not shown) and in the ocean interior (Fig. 10), and the low salinity water persists in the intermediate layer. To investigate the

cause of this salinity improvement, we estimate the salinity budget equation [Eq. (9)] difference between the AOEI and CTL runs (Fig. 11):

$$\Delta \left( \frac{\partial S}{\partial t} \right) = \Delta \{ \nabla (\boldsymbol{\kappa} \circ \nabla S) \} - \Delta \{ \boldsymbol{v} \cdot \nabla S \} + \Delta (\boldsymbol{increment}), \tag{10}$$

where $\Delta$ indicates the AOEI run minus the CTL run. The salinity tendency difference term [the LHS term of Eq. (10)] indicates that the salinity structure is maintained in the AOEI run by suppressing the salinity increase throughout the experiment period (Fig. 11a). The diffusion and advection difference terms [the first and second terms on the RHS of Eq. (10), respectively] contribute almost equally to the salinity tendency difference term. The diffusion difference term is dominated by only the vertical diffusion difference, whereas the advection difference term is dominated by different components in different months:

by the meridional advection difference in July; by all advection differences in August–September; and by vertical and partly zonal advection differences in October–December 2015. The reduction in the vertical diffusion is therefore the main cause of the improvement for low salinity water in the AOEI run relative to the CTL run. Figure 12 shows the vertical profile of the vertical diffusivity $\kappa_z$ averaged over the KE region (140°–160°E, 30°–40°N) for the whole experiment period and the maximum of the averaged diffusivity over the whole experiment period within 300–1000 m depth. As is consistent with the

results of the salinity budget analysis, there is exceedingly large vertical diffusivity at 300–800 m depth around the KE region, which results in salinity degradation induced by strong vertical diffusion in the CTL run. In contrast, the low salinity water in the intermediate layer persists in the AOEI run because the vertical diffusivity is smaller.



Weak density stratification and strong vertical shear are favorable conditions for the generation of large vertical diffusivity (Davis et al., 2016; Pacanowski and Philander, 1981). To gain dynamical insight into the vertical diffusivity

difference between the AOEI and CTL runs, the temporal tendency of the vertical diffusivity $\kappa_z$, the squared buoyancy frequency $N^2 = -g/\sigma_\theta\, \partial\sigma_\theta/\partial z$, and the squared vertical shear $\boldsymbol{u}_z^2 = |\partial\boldsymbol{u}/\partial z|^2$ ($\partial\kappa_z/\partial t$, $\partial N^2/\partial t$, and $\partial\boldsymbol{u}_z^2/\partial t$, respectively) are summed during the positive vertical diffusivity tendency ($\partial\kappa_z/\partial t > 0$), and are then averaged in the KE region (140°–160°E, 30°–40°N). The density vertical gradient can be decomposed as

$$\frac{\partial\sigma_\theta}{\partial z} = -\sigma_\theta\alpha_T\frac{\partial T}{\partial z} + \sigma_\theta\beta_S\frac{\partial S}{\partial z}, \tag{11}$$

where $\alpha_T$ ($\beta_S$) is the thermal (salinity) expansion coefficient. Consequently, the buoyancy frequency can be represented as the sum of the contributions from the temperature and salinity vertical gradients ($N_T^2$ and $N_S^2$, respectively):

$$N^2 = -\frac{g}{\sigma_\theta}\frac{\partial\sigma_\theta}{\partial z} = g\left(\alpha_T\frac{\partial T}{\partial z} - \beta_S\frac{\partial S}{\partial z}\right) \equiv N_T^2 + N_S^2. \tag{12}$$

Figure 13a, b shows that the total vertical diffusivity tendency is smaller in the AOEI run than the CTL run, which agrees qualitatively with the diffusivity averaged over the whole period (Fig. 12a, b). As is clear from Fig. 13c, d, the total shear tendency is almost zero in both the CTL and AOEI runs. The total buoyancy frequency tendency makes substantial

contributions in the CTL and AOEI runs, and its amplitude is smaller in the AOEI run than the CTL run. Since the negative values indicate weakening of the density stratification, the density stratification is less weakened in the AOEI run than the CTL run. The difference in the total buoyancy frequency tendency between the AOEI and CTL runs is caused by the differences of both the total $\partial N_T^2/\partial t$ and $\partial N_S^2/\partial t$ (Fig. 13e). We note that $\partial N_T^2/\partial t$ ($\partial N_S^2/\partial t$) can be decomposed into the temporal tendency terms of the vertical temperature (salinity) gradient and the thermal (salinity) expansion coefficient:


$$\frac{\partial N_T^2}{\partial t} = g\alpha_T\frac{\partial}{\partial t}\frac{\partial T}{\partial z} + g\frac{\partial T}{\partial z}\frac{\partial\alpha_T}{\partial t} \text{ and} \tag{13}$$

$$\frac{\partial N_S^2}{\partial t} = -g\beta_S\frac{\partial}{\partial t}\frac{\partial S}{\partial z} - g\frac{\partial S}{\partial z}\frac{\partial\beta_S}{\partial t}. \tag{14}$$

We confirmed that the latter terms have almost no impact on $\partial N_T^2/\partial t$ and $\partial N_S^2/\partial t$ (figure not shown). Therefore, the

differences in the temperature and salinity vertical gradient tendencies result in less weakening of the density stratification in the AOEI run than the CTL run.

To investigate the causes of the differences in the temperature and salinity vertical gradient tendencies between the AOEI and CTL runs, we take the vertical derivatives of the temperature and salinity budget equations in the ocean interior to obtain the temperature and salinity stratification tendency equations:






$$\frac{\partial}{\partial t}\left(\frac{\partial T}{\partial z}\right) = \frac{\partial}{\partial z}\{\boldsymbol{\nabla}(\boldsymbol{\kappa} \circ \boldsymbol{\nabla}T)\} - \frac{\partial}{\partial z}(\boldsymbol{v} \cdot \boldsymbol{\nabla}T) + \frac{1}{\rho_0 c_p}\frac{\partial q_{sw}}{\partial z} + \frac{\partial}{\partial z}(increment) \text{ and} \tag{15}$$

$$\frac{\partial}{\partial t}\left(\frac{\partial S}{\partial z}\right) = \frac{\partial}{\partial z}\{\boldsymbol{\nabla}(\boldsymbol{\kappa} \circ \boldsymbol{\nabla}S)\} - \frac{\partial}{\partial z}(\boldsymbol{v} \cdot \boldsymbol{\nabla}S) + \frac{\partial}{\partial z}(increment), \tag{16}$$

respectively. Here, $\rho_0 = 1025 \; kg \; m^{-3}$ is the reference density, $c_p = 4190 \; J \; kg^{-1} \; °C^{-1}$ is the specific heat of the seawater,

and $q_{sw}$ is downward shortwave radiation parameterized by

$$q_{sw} = Q_{sw}\left\{Rexp\left(-\frac{|z|}{\gamma_1}\right) + (1-R)exp\left(-\frac{|z|}{\gamma_2}\right)\right\} \tag{17}$$

(Paulson and Simpson, 1977), where $Q_{sw}$ is shortwave radiation at the sea surface, $R = 0.62$ is a separation constant, and $\gamma_1 =$

$0.60 \; m$ and $\gamma_2 = 20.0 \; m$ are attenuation length scales. These values are set to the case of Type IA from Jerlov (1976). As in
the total vertical diffusivity tendency calculated the above, the terms in Eqs. (15) and (16) are summed when $\partial \kappa_z / \partial t > 0$, and
then averaged in the KE region (140°–160°E, 30°–40°N) (Fig. 14). We note that positive values in Eqs. (15) and (16) indicate
opposite effects on the density stratification: a positive temperature (salinity) vertical gradient tendency strengthens (weakens)
the density stratification.

310         In the CTL and AOEI runs, the temperature gradient tendency term [the LHS term of Eq. (15)] is negative and
indicates that the temperature and density stratification are weakened at all depths (Fig. 14a, b). Compared with the CTL run,
the amplitude of this term is smaller (Fig. 14a, b), and thus the temperature and density stratification is less weakened in the
AOEI run. As shown in Fig. 14c, the difference in the temperature gradient tendency between the AOEI and CTL runs is due
mainly to the analysis increment gradient term [the last term on the RHS of Eq. (15)] and in part to the advection gradient term

[the second term on the RHS of Eq. (15)], whereas the diffusion and shortwave penetration gradient terms [the first and third
terms on the RHS of Eq. (15), respectively] make almost no contribution.

In the CTL run, the salinity gradient tendency term [the LHS term of Eq. (16)] indicates that the salinity (density)
stratification is strengthened (weakened) at all depths (Fig. 14d). In the AOEI run, the salinity (density) stratification is
weakened (strengthened) at 200–400 m depth and slightly strengthened (weakened) at 400–1000 m depth (Fig. 14e). The

salinity gradient tendency term is smaller in the AOEI run than the CTL run at all depths, and thus the salinity (density)
stratification is less strengthened (weakened) in the AOEI run relative to the CTL run (Fig. 14f). The difference in the analysis
increment gradient terms [the last term on the RHS of Eq. (16)] between the AOEI and CTL runs dominates that in the salinity
gradient tendency term, whereas the differences between the diffusion and advection gradient terms [the second and third
terms on the RHS of Eq. (14), respectively] have little influence. This indicates that less strengthening (weakening) of the

salinity (density) stratification in the AOEI run relative to the CTL run is due to the smaller analysis increment in the AOEI
run. The impacts of the SST, SSS, and SSH assimilation are limited to between the surface and about 370 m depth because of





the prescribed vertical localization scale of 100 m described in subsection 2.2, and consequently only in-situ temperature and salinity assimilation generates analysis increments in the intermediate layer.

The AOEI contributes to maintaining the density stratification by reducing the temperature and salinity increments
and preventing the occurrence of large vertical diffusivity that degrades low salinity water in the intermediate layer around the KE region. In the CTL run, the salinity increments restore the degraded low salinity water (Fig. 5) but lead to degradation through the formation of large vertical diffusivity at the same time. Thus, it seems that a positive feedback exists that may degrade the salinity structure.

### 3.4 Improvement of geostrophic balance and accuracy

In this section, we investigate the impacts of the AOEI on the geostrophic balance and accuracy. Figure 15 shows $\Delta NBE$ averaged over the whole period in the CTL and AOEI runs. In the CTL run, $\Delta NBE$ is large in the mid-latitude regions, especially along the KE (Fig. 15a). In the AOEI run, $\Delta NBE$ is smaller than the CTL run for the entire domain (Fig. 15b). The spatiotemporal averaged $\Delta NBE$ over the whole experiment period and domain is $0.57 \times 10^{-10}$ s$^{-2}$ and $0.35 \times 10^{-10}$ s$^{-2}$ for the
CTL and AOEI runs, respectively, and the balance is significantly improved in the AOEI run relative to the CTL run. This is probably because the analysis increments are smaller in the AOEI run than the CTL run.

To investigate the accuracy in the ocean interior, we calculate the RMSDs relative to in-situ temperature, salinity, and horizontal velocity observations from the KEO buoy south of the KE (Figs. 15a, 16). Results are only presented for the temperature and salinity because no significant results are obtained for the horizontal velocity. The RMSDs for both
temperature and salinity are smaller in the AOEI run than the CTL run, and the AOEI run provides significant temperature (salinity) improvements at 0–150 m (50–400 m) depth relative to the CTL run. This is probably because the AOEI suppresses the development of the strong vertical diffusion that leads to the salinity degradation and because of the improvement in the balance.

We also investigate the accuracy of the surface flow field, calculating the spatiotemporally averaged RMSDs relative
to the SSH and SSHA datasets from the AVISO, and to in-situ surface horizontal velocity observations from the drifter buoys (Fig. 17). The RMSDs are smaller for all variables in the AOEI run, and indicate significant improvements relative to the CTL run, with the exception of surface meridional velocity. The SSHA RMSDs averaged over the experiment period also show that the AOEI run may represent an improvement relative to the CTL run in some high RMSD regions; for example, along the Kuroshio (130–140°E, 30°–35°N) and downstream of the KE (160°–165°E, 30°N) (Fig. 17a, b). Here, we note that these
improvements are not necessarily caused by the larger ensemble spread in the AOEI run than the CTL run, as the AOEI decreases the analysis increments by inflating the observation errors. Instead, the better balance and accuracy of the density structure in the ocean interior in the AOEI run would result in an improvement in the surface flow field.

### 4. Summary



We have implemented the AOEI with the sbPOM-LETKF ocean data assimilation system and conducted sensitivity experiments to investigate the impacts on the low-salinity NPIW around the KE region, geostrophic balance, and accuracy in the analysis field. In the CTL run, the large analysis increments by in-situ temperature and salinity assimilation weaken the density stratification. The resulting exceedingly large vertical diffusivity induces the strong vertical diffusion that breaks the low salinity structure in the NPIW around the KE region. The salinity increment contributes to restoring the low salinity water,

but at the same time causes the salinity degradation by generating strong vertical diffusion. Therefore, the positive feedback appears to occur, degrading the salinity structure.

The AOEI decreases the temperature and salinity increments around the KE region by adaptively inflating the temperature and salinity observation errors, respectively. As a result, the AOEI mitigates the salinity degradation seen in the CTL run, and therefore, the low salinity water is maintained in the AOEI run. In addition, the AOEI significantly improves the

geostrophic balance probably because of the reduction of the analysis increments. Moreover, the AOEI prevents the development of strong vertical diffusion and improves the accuracy of temperature and salinity in the ocean interior. Furthermore, the improvements of the balance and density structure in the ocean interior contribute to more accurate surface flow field. In summary, this study demonstrated the positive impacts of the AOEI on the balance and accuracy of the temperature, salinity, and surface flow fields.

Since our available computational resources were limited, we fixed the tuning parameter of the RTPP, perturbed atmospheric forcing, ensemble size, localization scale, and prescribed observation errors. Further experiments to explore more optimal settings are required and will be investigated in the future. Since low salinity water is distributed in the intermediate layer in western boundary current regions in all ocean basins, we would expect that this study will be helpful for improving existing EnKF-based ocean data assimilation systems. Minamide and Zhang (2017) noted that the AOEI has the advantage of

being easily implemented with various EnKF-based systems, and this study serves as a good example for the usefulness of the AOEI. We are currently constructing high-resolution reanalysis datasets in the western North Pacific and Maritime Continent regions based on this system (Ohishi et al. in prep.), and plan to develop (near) real-time ensemble forecast systems.

**Code and data availability**

The source code for sbPOM version 1.0 and LETKF are available from https://github.com/shunohishi/sbPOM-LETKF (last access: 29 March 2022, Jordi and Wang, 2012; Ohishi et al. in review) and https://github.com/takemasa-miyoshi/letkf (last access: 13 April 2021, Miyoshi and Yamane, 2007), respectively. The source code for COARE version 3.5 (Brodeau et al., 2017; Edson et al., 2013) was downloaded from https://github.com/brodeau/aerobulk (last access: 13 April 2021).

We thank Dr. Kenshi Hibino for providing us with an earlier version of TE-Global, before the official release of the latest version (https://www.eorc.jaxa.jp/water/, last access: 13 April 2021). The observation datasets are: the surface drifter buoy data (https://www.aoml.noaa.gov/phod/gdp/hourly_data.php, last access: 13 April 2021, Elipot et al., 2016); the KEO buoy data (https://www.pmel.noaa.gov/ocs/, last access: 13 April 2021); ETOPO1 (https://www.ngdc.noaa.gov/mgg/global/,



last access: 13 April 2021, Amante and Eakins, 2009); WOA18 (https://www.ncei.noaa.gov/access/world-ocean-atlas-2018/,
last access: 13 April 2021; Locarnini et al., 2019; Zweng et al., 2019); the satellite SSTs from Himawari-8
(https://www.eorc.jaxa.jp/ptree/index.html, last access: 13 April 2021; Bessho et al., 2016; Kurihara et al., 2016) and GCOM-
W (https://gportal.jaxa.jp/gpr/?lang=en, last access: 13 April 2021); the satellite-derived SSS from SMOS
(http://www.esa.int/Applications/Observing_the_Earth/SMOS, last access: 13 April 2021) and SMAP version 4.3
(https://podaac.jpl.nasa.gov/, last access: 13 April 2021, Meissner et al., 2018); the satellite-derived SSHA and AVISO (Ducet
et al., 2000) from CMEMS (https://marine.copernicus.eu/, last access: 13 April 2021); and in-situ temperatures and salinity
from GTSPP (https://www.ncei.noaa.gov/products/global-temperature-and-salinity-profile-programme, last access: 13 April
2021, Sun et al., 2010) and AQC Argo version 1.2a (http://www.jamstec.go.jp/ARGO/argo_web/argo/?page_id=100&lang=en,
last access: 13 April 2021). The global JRA55 atmosphere and SODA 3.7.2 ocean reanalysis datasets are from
http://search.diasjp.net/en/dataset/JRA55 (last access: 13 April 2021, Kobayashi et al., 2015) and
https://www.soda.umd.edu/soda3_readme.htm (last access: 13 April 2021, Carton et al., 2018), respectively.

**Author contribution**

SO developed the code for the ocean data assimilation system, conducted the sensitivity experiments, and analyzed the outputs.
SO and TM prepared the paper with contributions from MK.


**Competing interests**

The authors declare that they have no conflict of interest.

**Acknowledgements**

Numerous comments from Drs. Nariaki Hirose, Takahiro Toyoda, Yosuke Fujii, and Norihisa Usui at the Meteorological
Research Institute, Yoichi Ishikawa at JAMSTEC, Katsumi Takayama at IDEA Consultants, Inc., Naoki Hirose at Kyushu
University, and participants in the ocean data assimilation summer school helped us to develop the presented system. This
work used computational resources of the JAXA Supercomputer System Generation 2 and 3 (JSS2 and JSS3, respectively)
and the supercomputer Fugaku provided by RIKEN through the HPCI Research Project (Project ID: hp210166, ra000007).


**Financial support**

This work was supported by JST AIP Grant Number JPMJCR19U2, Japan; MEXT (JPMXP1020200305) as "Program for
Promoting Researches on the Supercomputer Fugaku" (Large Ensemble Atmospheric and Environmental Prediction for
Disaster Prevention and Mitigation); the COE research grant in computational science from Hyogo Prefecture and Kobe City
through Foundation for Computational Science; JST, SICORP Grant Number JPMJSC1804, Japan; JSPS KAKENHI Grant
Number JP19H05605; the JAXA; JST, CREST Grant Number JPMJCR20F2, Japan; Cabinet Office, Government of Japan,
Cross-ministerial Moonshot Agriculture, Forestry and Fisheries Research and Development Program, "Technologies for Smart



Bio-industry and Agriculture"(funding agency: Bio-oriented Technology Research Advancement Institution) No. JPJ009237; RIKEN Pioneering Project "Prediction for Science"; RIKEN Engineering Network Project.


**Review statement**

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





Figure 1: Monthly-mean SST for (a) July, (c) October, and (e) December 2015 in the CTL run. (b), (d), and (f) Same as in (a), (c), and (e), but for assimilated satellite SSTs. Thin (thick) black contour intervals are 2 (10) °C. White lines in (a) indicate 150°E (35°N) used for the zonal (meridional) sections in Fig. 3 (4).


**Figure 2: Same as Fig.1 but for SSS. Thin (thick) black contour intervals are 0.25 (2). In (b), (d), and (f), contour intervals are not shown because the satellite observations are noisy.**



**Figure 3: The meridional section of monthly-mean temperature (color) and potential density (contour) along 150°E in (a) July, (c) October, and (e) December 2015 in the CTL run. (b), (d), and (f) Same as (a), (c), and (e), but for salinity. Thin (thick) contour intervals are 0.25 (2) kg m$^{-3}$. The white box in (f) encloses a latitude-depth section of 30°–40°N and 500–1000 m depth.**



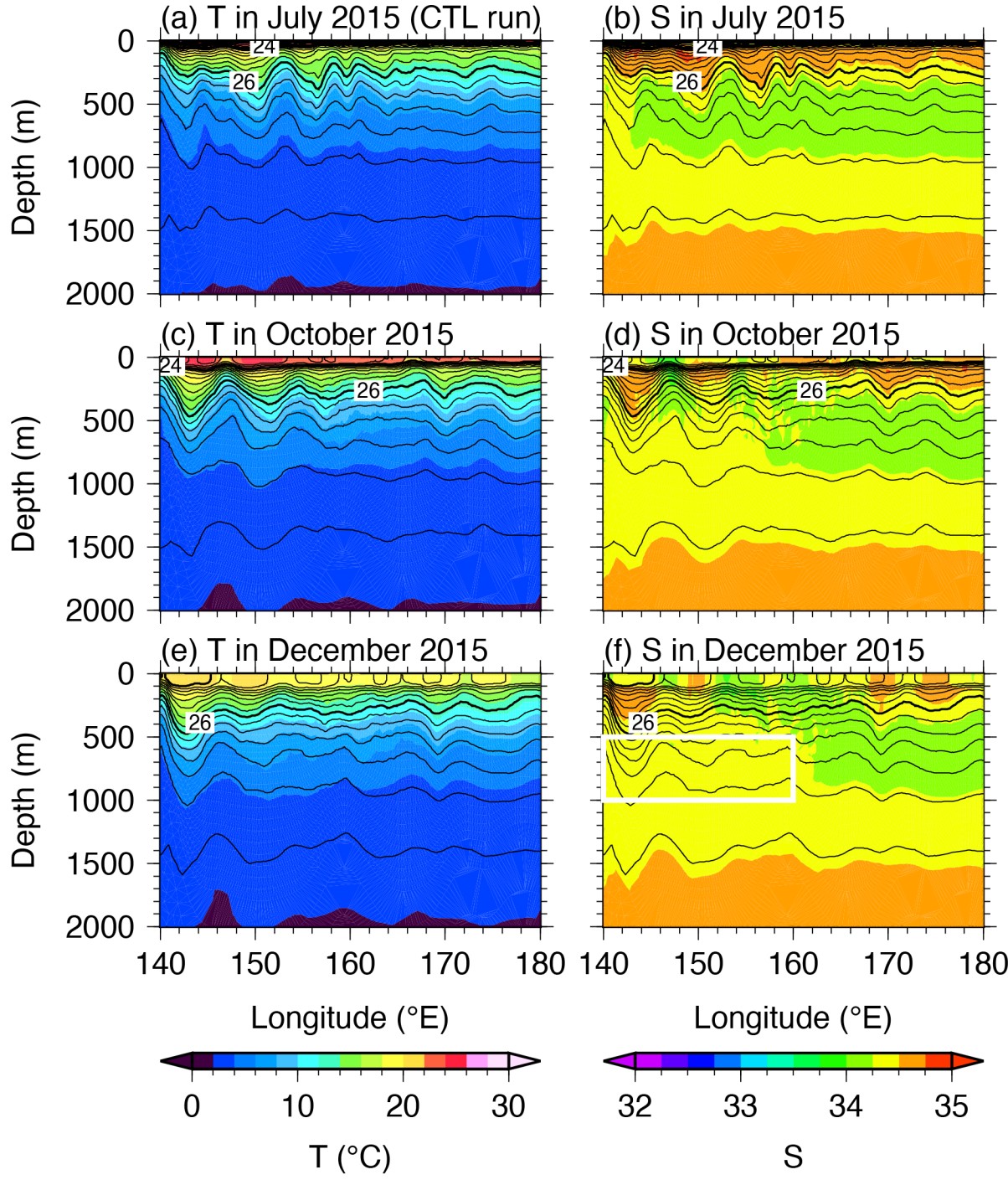

Figure 4: Same as Fig. 3, but for the zonal section along 35°N. The white box in (f) encloses a depth longitude-depth section of 140°–160°E and 500–1000 m.



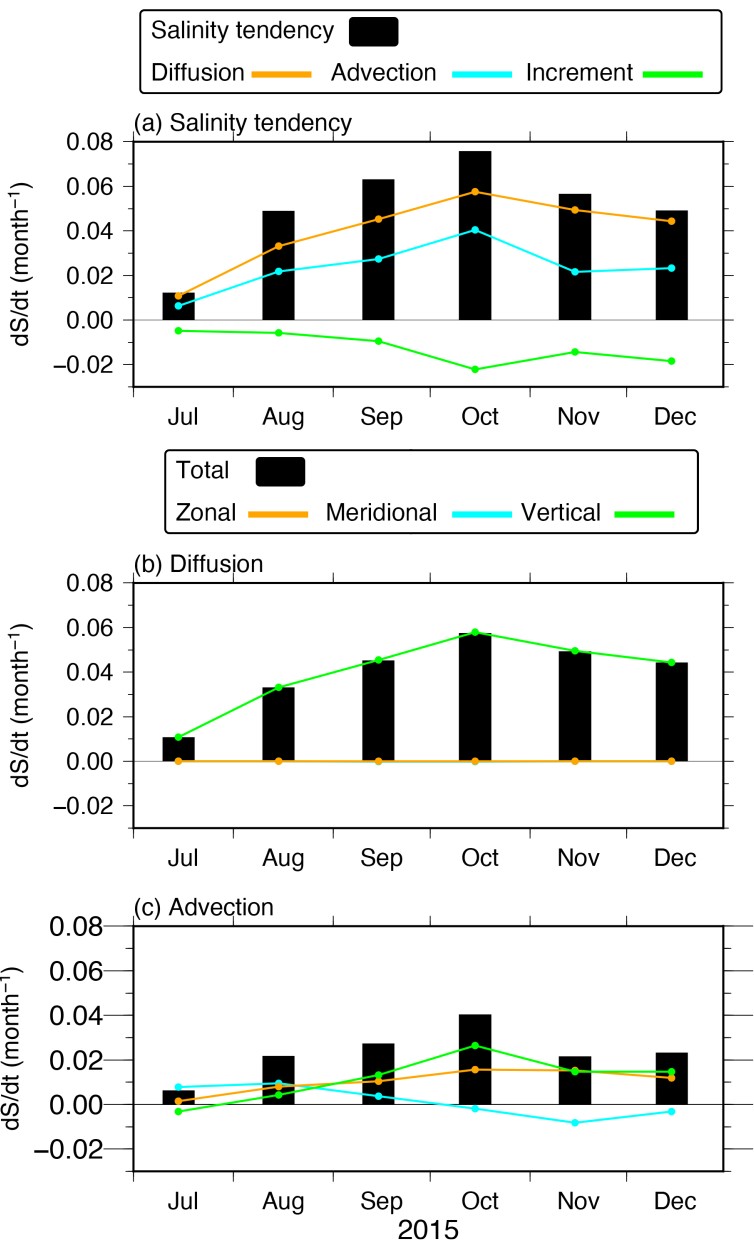

**Figure 5: (a) Monthly mean for each term in the salinity budget equation [Eq. (9)] averaged over the KE region in the intermediate layer (140°–160°E, 30°–40°N, and 500–1000 m depth) in the CTL run: the salinity tendency term (the LHS term; black bars), salinity diffusion term (the first term on the RHS; orange line), salinity advection term (the second term on the RHS, cyan line), and salinity increment term (the last term on the RHS; green line). (b) and (c) Same as (a), but for salinity diffusion and advection terms (black bars), and zonal (orange lines), meridional (cyan lines), and vertical (green lines) components.**



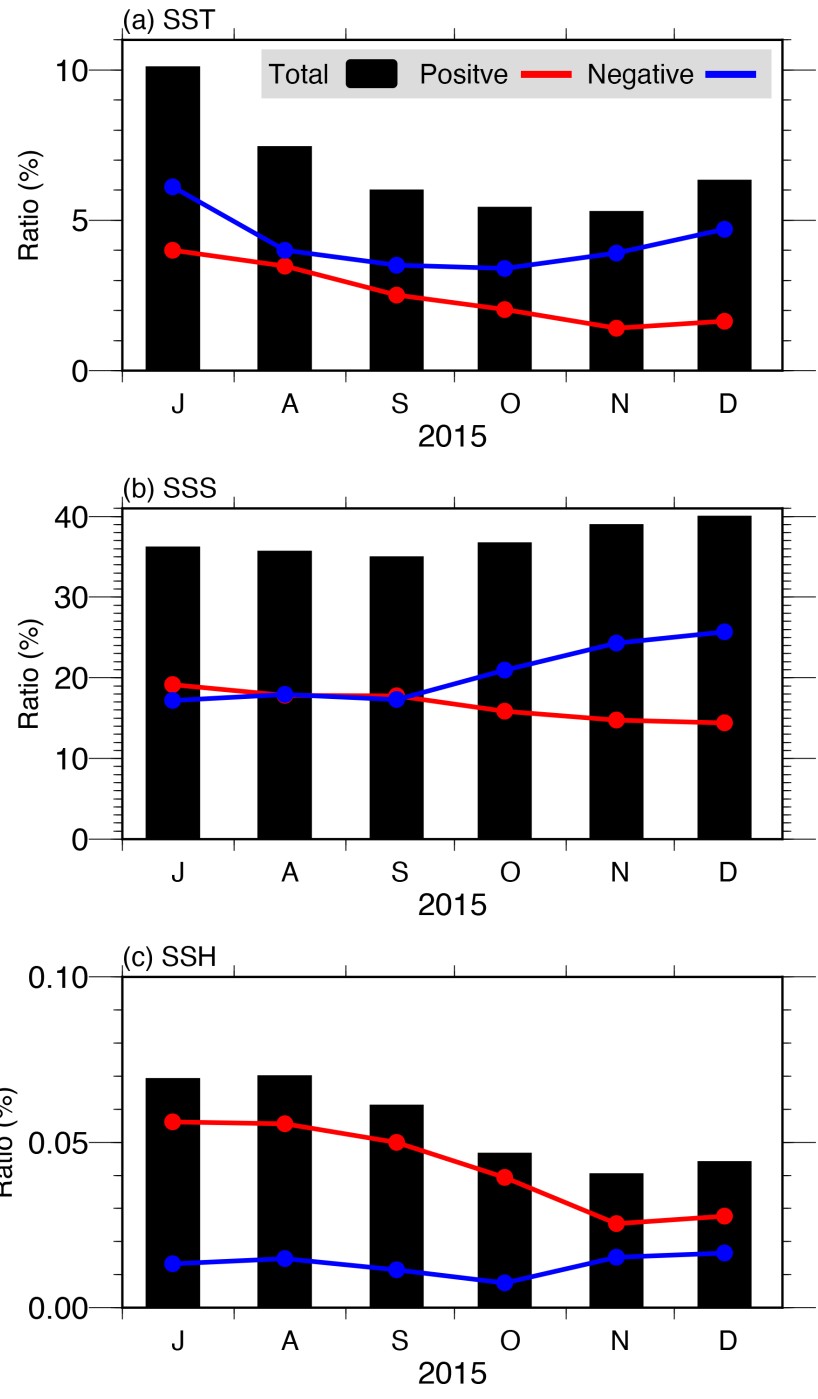

**Figure 6: Monthly-mean ratio of the area where the AOEI is applied to the system domain in the (a) SST, (b) SSS, and (c) SSH fields in the AOEI run (black bars). Red (blue) lines indicate when the innovation is positive (negative).**





**Figure 7: (a) Ratio of the period when the AOEI is applied to the SST compared with the whole experiment period in the AOEI run. (b) and (c) Same as (a) but for when the innovation is positive and negative, respectively. (d) and (e) innovation amplitude and ensemble spread averaged during the period when the AOEI is applied, respectively. Black contours indicate SST averaged over the whole period. Thin (thick) contour intervals are 2 (10) °C.**

**Figure 8: Same as Fig. 7 but for SSS. Thin (thick) contour intervals are 0.25 (1).**






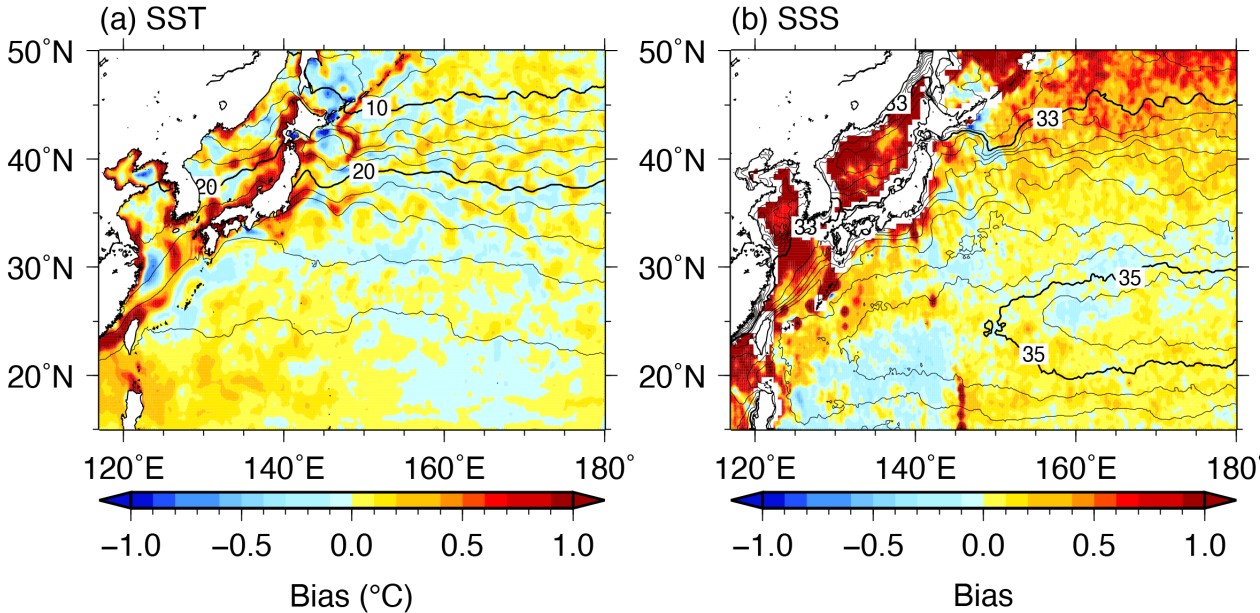

Figure 9: (a) SST and (b) SSS forecast biases (color) and averages (contour) over the whole experiment period. Thin
(thick) contour intervals are 2 (10) °C in (a) and 0.25 (2) in (b).

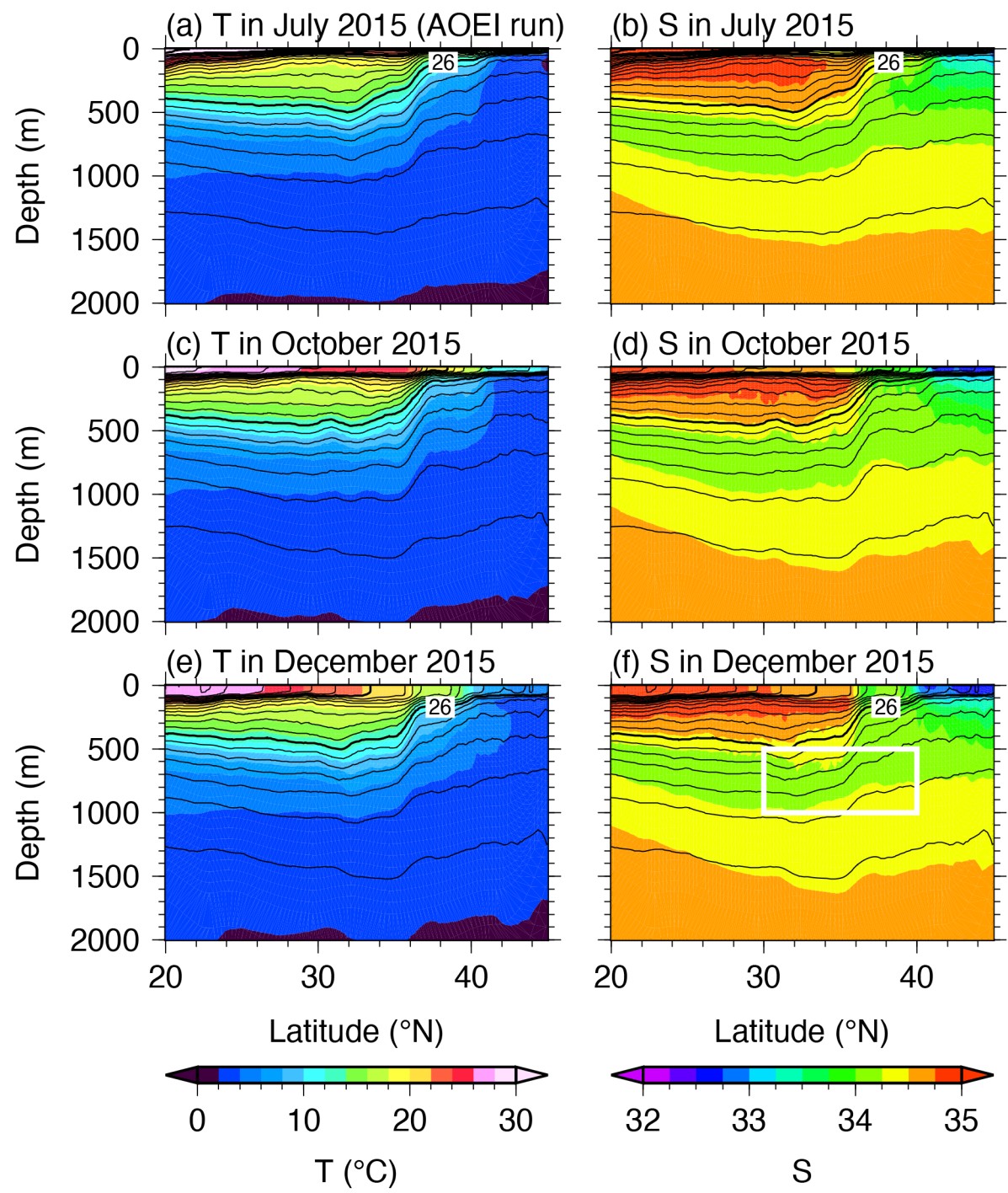

**Figure 10: Same as Fig. 3 but for the AOEI run.**



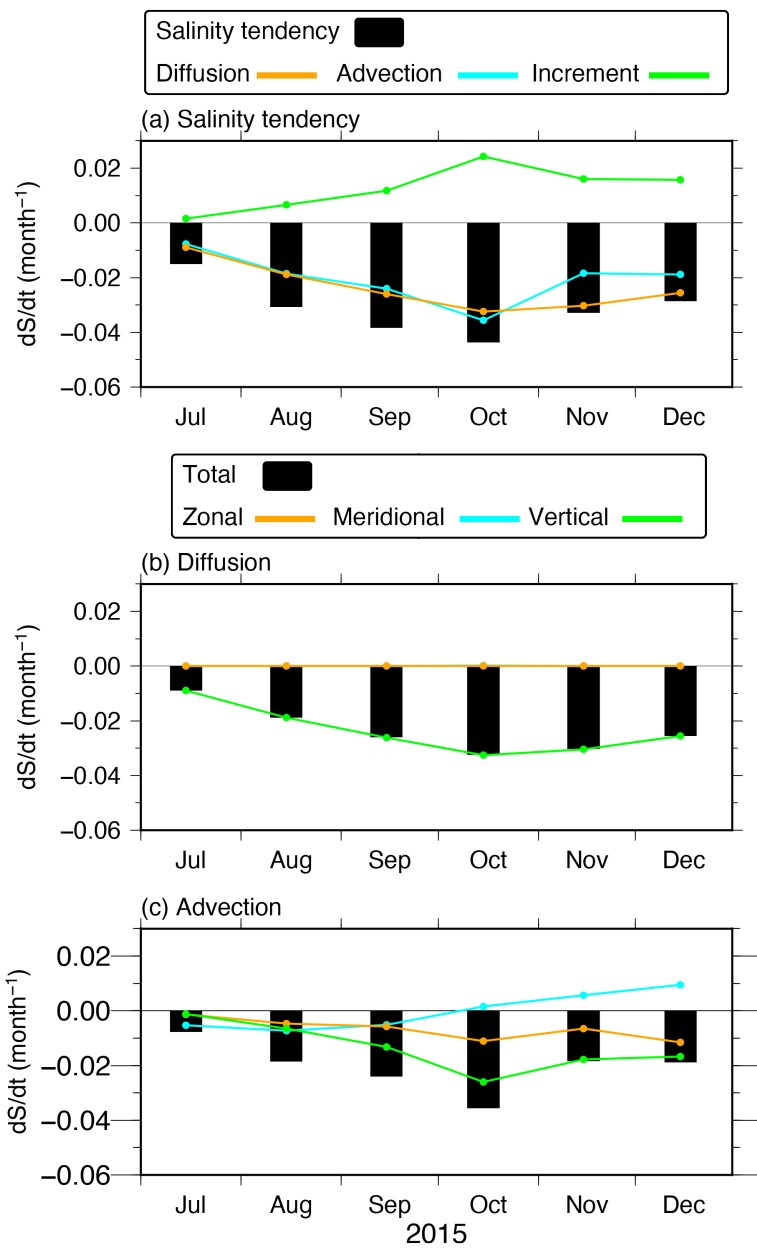


**Figure 11: (a) Monthly mean for each term for the difference between the AOEI and CTL runs in the salinity budget equation [Eq. (10)]: Salinity tendency difference term (the LHS term; black bars), salinity diffusion difference term (the first term on the RHS; orange line), salinity advection difference term (the second term on the RHS; cyan line), and salinity increment difference term (the last term on the RHS; green line). (b) and (c) Same as (a) but for the salinity**

**diffusion and advection difference terms (black bars), respectively, and zonal (orange lines), meridional (cyan lines), and vertical (green lines) components.**





**Figure 12: Vertical diffusivity (black lines), and squared buoyancy frequency (red lines) and shear (blue lines) averaged over the KE region (140°–160°E, 30°–40°N) for the whole experiment period in the (a) CTL and (b) AOEI runs. Maxima of the averaged vertical diffusivity for the whole experiment period within 300–1000 m depth in the (c) CTL and (d) AOEI runs. Black contours show SSH averaged over the whole period. Thin (thick) contour intervals are 0.2 (1) m.**



**Figure 13: Total vertical diffusivity tendency during the positive vertical diffusivity tendency period averaged over the KE region (140°–160°E, 30°–40°N) in the (a) CTL and (b) AOEI runs. (c) and (d) Same as (a) but for the squared buoyancy frequency (black) and shear (gray) tendency. (e) Same as (c) and (d), but for the AOEI minus CTL run. In (c)–(e), cyan (orange) lines indicate contributions from $\partial N_T^2/\partial t$ ($\partial N_S^2/\partial t$).**







**Figure 14:** (a)–(c) Same as Fig. 13 but for each term of the temperature stratification budget equation [Eq. (15)]: Temperature gradient tendency term (the LHS term; black), temperature diffusion gradient term (the first term on the RHS; red), temperature advection gradient term (the second term on the RHS; blue), shortwave penetration gradient term (the third term on the RHS; orange), and temperature increment gradient term (the last term on the RHS; cyan). The shortwave penetration gradient term is almost zero and overlaps with the temperature diffusion gradient term. (d)–(f) Same as (a)–(c) but for the salinity stratification budget equation [Eq. (16)].



**Figure 15:** Δ*NBE* (color) and SSH (contour) averaged over the whole experiment period for the (a) CTL and (b) AOEI runs. Thin (thick) contour intervals are 0.2 (1) m. Spatiotemporally averaged Δ*NBE* over the whole period and domain is shown in the lower right corners. The black star in (a) denotes the position of the KEO buoy.





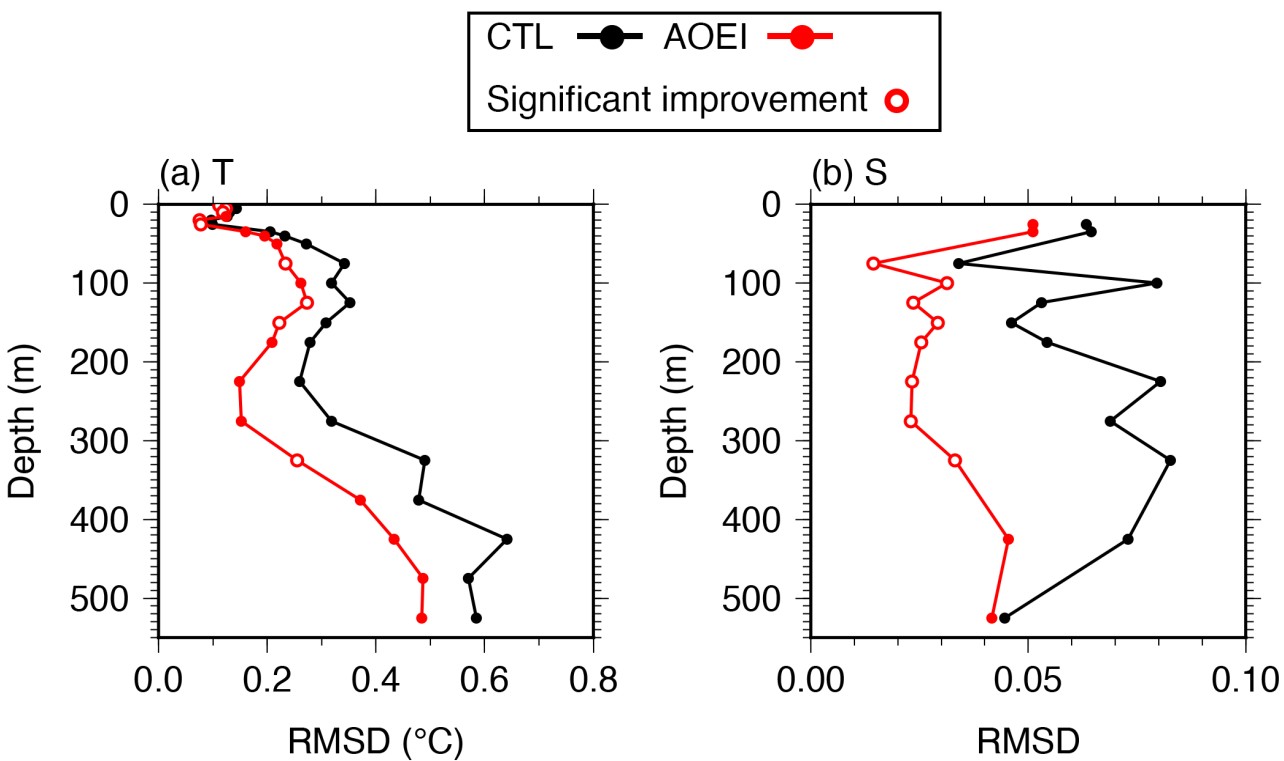

Figure 16: (a) Temperature and (b) salinity RMSDs averaged over the whole experiment period at the KEO buoy in the CTL (black) and AOEI (red) runs. Open circles indicate significant improvement in the AOEI run relative to the

645  CTL run.





Figure 17: RMSDs for (a) SSH and (b) SSHA relative to the AVISO, and for surface (c) zonal and (d) meridional velocity relative to the drifter buoys over the whole domain and period. Black dots indicate the ensemble spread in the observation space.



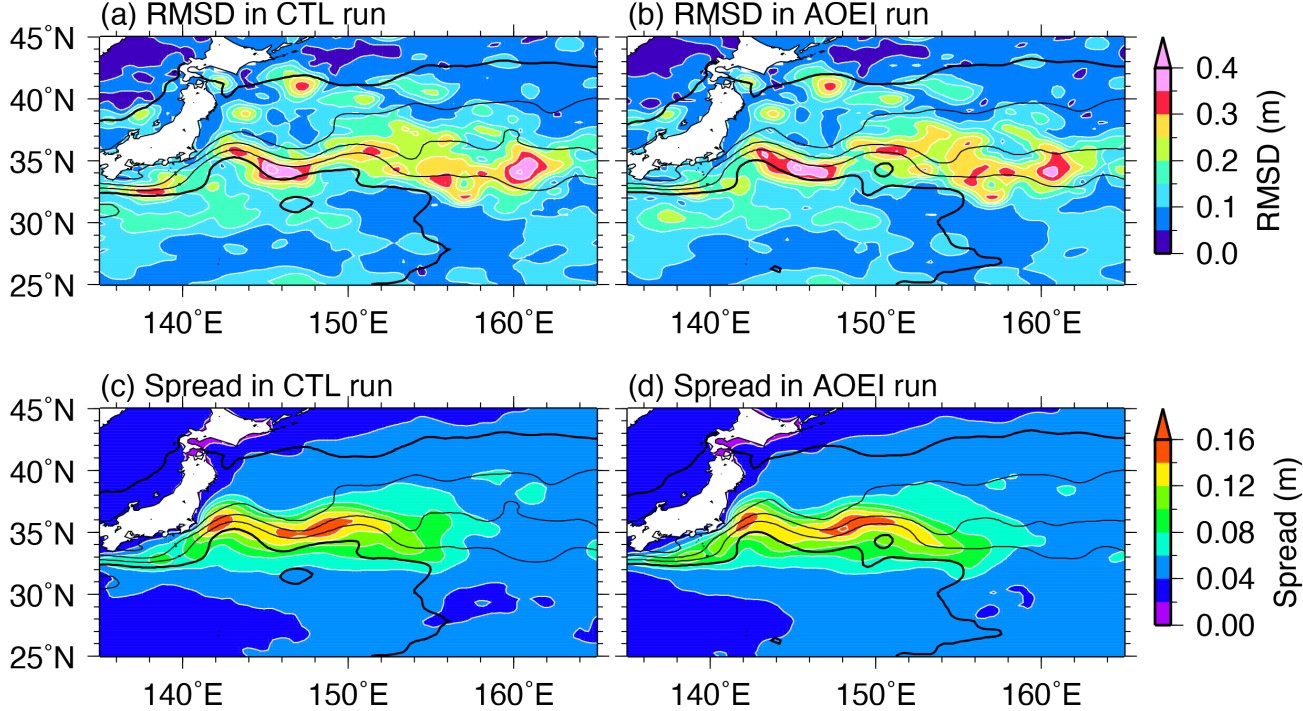

**Figure 18: RMSDs for SSHA relative to the AVISO averaged over the whole experiment period in the (a) CTL and (b) AOEI runs. (c) and (d) Same as (a) and (b) but for the SSHA ensemble spread.**

655