# Peer review of "An EnKF-based ocean data assimilation system improved by adaptive observation error inflation (AOEI)"

_Geoscientific Model Development, 2022_

## Referee Comment (RC1)

An EnKF-based ocean data assimilation system improved by adaptive observation error inflation (AOEI)

This work addresses the general issue of taking best advantage from dense and high-resolution satellite observations, in particular satellite sea surface salinity (SSS) data, to improve our prediction and knowledge of ocean dynamics. One main difficulty relates to the large discrepancies between observation and forecast ensembles that can appear in frontal regions due to mismatches of the sky type (clear versus cloudy). These large discrepancies may induce unphysical analysis corrections in frontal regions. To overcome this issue, the objective of this study is to adapt the AOEI method (adaptive observation error inflation based on Desroziers' innovation diagnostics) to an EnKF-based ocean data assimilation (3D-LETKF formulation with 100 members). The AOEI provides thereby a way to inflate observations errors with a spatial dependency.

In this paper, the Authors study the AOEI impact on salinity structure, geostrophic balance and accuracy in the northwestern Pacific region when all-sky infrared brightness temperatures are assimilated at one-day time intervals. They illustrate the degradation of the salinity structure resulting from EnKF analysis without AOEI and impacting vertical diffusion. They also demonstrate that including AOEI within the EnKF can successfully limit the erroneous analysis increments and thereby preserve the salinity structure.

I find that the paper is generally well written and clear. The issue raised in this work is of great interest for the geoscience community because it is essential to be able to take advantage of current and future satellite observations to improve model predictions. The case study and results are relevant to answer this question.

I also find that the introduction shall put more emphasis on the objectives and on the issues associated with data assimilation when dealing with structures/patterns and therefore position errors. This is a general problem of standard data assimilation algorithms, which have been designed to handle amplitude errors and not position errors. The AOEI method provides a way to limit the issues of position errors in areas where observation errors may also be large. However, it would be of interest to readers to replace this issue in the more general context of position error treatment in data assimilation systems. Adding some comments in the introduction and conclusion on this aspect would be worthwhile.

By the way, I find that the paragraph "As shown in section 3, an EnKF-based ocean data assimilation system… are large due to fronts and eddies." (l. 69-72) is not at the right place in the introduction. It is surprising to announce in quite significant details the results found in the paper directly in the introduction. I suggest the Authors to modify/reformulate this part to further discuss the idea of position errors.

Also, in lines 325-329, there is a discussion on the limits of the SST, SSS and SSH assimilation due to the prescribed vertical localization scale. It is not clear in the text if this limit is satisfying or if there is some work to be done to overcome this limitation. How was this localization scale defined? How does it impact the vertical diffusion processes discussed in the paper? Comment on this aspect would be valuable.

**(*) Some additional minor comments**
I encourage the Authors to
- Throughout the manuscript, change "1 day" to "one day"
- Throughout the manuscript, write "Section" with a capital letter at the beginning of sentences, and use the abbreviation "Sect." within sentences.
- l. 15, correct typographical error "by combining forecasts and observations"
- l. 40, add references related to variational approaches for ocean data assimilation
- l. 42, correct grammar error "provide a large number"

- l. 147, remove the reference Ohishi et al. (in preparation): it is not conventional to cite a paper that is in preparation, it should be at least accessible in some ways.
- l. 169, remove the word "taking"
- l. 171, precise what is meant by "the accuracy" (the accuracy of what?)
- l. 350, define the acronym SSHA the first time it appears in the text
- Bibliography, modify the year for reference by Desroziers et al. (2005)

---

## Author Response (AR1)

This work addresses the general issue of taking best advantage from dense and high-resolution satellite observations, in particular satellite sea surface salinity (SSS) data, to improve our prediction and knowledge of ocean dynamics. One main difficulty relates to the large discrepancies between observation and forecast ensembles that can appear in frontal regions due to mismatches of the sky type (clear versus cloudy). These large discrepancies may induce unphysical analysis corrections in frontal regions. To overcome this issue, the objective of this study is to adapt the AOEI method (adaptive observation error inflation based on Desroziers' innovation diagnostics) to an EnKF-based ocean data assimilation (3D-LETKF formulation with 100 members). The AOEI provides thereby a way to inflate observations errors with a spatial dependency. In this paper, the Authors study the AOEI impact on salinity structure, geostrophic balance and accuracy in the northwestern Pacific region when all-sky infrared brightness temperatures are assimilated at one-day time intervals. They illustrate the degradation of the salinity structure resulting from EnKF analysis without AOEI and impacting vertical diffusion. They also demonstrate that including AOEI within the EnKF can successfully limit the erroneous analysis increments and thereby preserve the salinity structure.

I find that the paper is generally well written and clear. The issue raised in this work is of great interest for the geoscience community because it is essential to be able to take advantage of current and future satellite observations to improve model predictions. The case study and results are relevant to answer this question. I also find that the introduction shall put more emphasis on the objectives and on the issues associated with data assimilation when dealing with structures/patterns and therefore position errors. This is a general problem of standard data assimilation algorithms, which have been designed to handle amplitude errors and not position errors. The AOEI method provides a way to limit the issues of position errors in areas where observation errors may also be large. However, it would be of interest to readers to replace this issue in the more general context of position error treatment in data assimilation systems. Adding some comments in the introduction and conclusion on this aspect would be worthwhile.

We thank the reviewer for insightful comments, especially on the position error. We have added the description of the differences in the position of the boundary between forecasts and observations to the third paragraph in Sect. 1.

By the way, I find that the paragraph "As shown in section 3, an EnKF-based ocean data assimilation system... are large due to fronts and eddies." (l. 69-72) is not at the right place

in the introduction. It is surprising to announce in quite significant details the results found in the paper directly in the introduction. I suggest the Authors to modify/reformulate this part to further discuss the idea of position errors. Also, in lines 325-329, there is a discussion on the limits of the SST, SSS and SSH assimilation due to the prescribed vertical localization scale. It is not clear in the text if this limit is satisfying or if there is some work to be done to overcome this limitation. How was this localization scale defined? How does it impact the vertical diffusion processes discussed in the paper? Comment on this aspect would be valuable.

In the third paragraph in Sect. 1, we have removed the description of the salinity degradation seen in the experiment, and added the general insight on the similarities of ocean fronts to the atmospheric boundaries between clear- and cloudy-sky, following the reviewer's comments.

We have conducted preliminary experiments with and without the vertical localization scale. We have found that the low-salinity structure is more likely broken in the experiment without vertical localization than that with vertical localization, probably because assimilating surface observations results in the larger analysis increments throughout the depth and causes the degradation mechanism. Therefore, we have set the vertical localization scale of 100 m following Miyazawa et al. (2012) and Penny et al. (2013). As described in the last paragraph in Sect. 4 in the original and revised manuscripts, the horizontal and vertical localization scales are not optimally tuned in this study, and this is an issue in future studies.

**(*) Some additional minor comments**

We thank the reviewer for checking carefully throughout the manuscript. We have modified corresponding parts following your comments.

I encourage the Authors to
- Throughout the manuscript, change "1 day" to "one day"

We have replaced "1 day" with "one day" in the Abstract, the third paragraph in Sect. 1, and the second paragraph in subsection 2.2.

- Throughout the manuscript, write "Section" with a capital letter at the beginning of sentences, and use the abbreviation "Sect." within sentences.

We have replaced "section" with "Sect." in the last paragraph in Sect. 1 and in the last sentence in subsection 2.1.

- l. 15, correct typographical error "by combining forecasts and observations"

We have replaced "bv" with "by" in the first sentence in the first paragraph in Sect. 1.

- l. 40, add references related to variational approaches for ocean data assimilation

At the end of the first paragraph in Sect. 1, we have added the citation of Miyazawa et al. (2017) and Zuo et al. (2019) in which 3D-VAR is adopted in ocean data assimilation systems.

- l. 42, correct grammar error "provide a large number"

We have replaced "provides" with "provide" in the second paragraph in Sect. 1.

- l. 147, remove the reference Ohishi et al. (in preparation): it is not conventional to cite a paper that is in preparation, it should be at least accessible in some ways.

In the first sentence in subsection 2.3, we have incorrectly cited Ohishi et al. (in preparation), and replaced "Ohishi et al. (in preparation)" with "Ohishi et al. (in review)". In the last sentence of Sect. 4, we have removed "Ohishi et al. (in prep.)".

- l. 169, remove the word "taking"

We have removed "taking" between "By" and "$\partial/\partial x$" in the second sentence in subsection 2.3.1.

- l. 171, precise what is meant by "the accuracy" (the accuracy of what?)

We have added "of temperature, salinity, horizontal velocities, and SSH" after "the accuracy" in the first sentence of subsection 2.3.2.

- l. 350, define the acronym SSHA the first time it appears in the text

We have added "(SSHAs)" after "sea surface height (SSH) anomalies" in the last sentence of the second paragraph in Sect. 1.

- Bibliography, modify the year for reference by Desroziers et al. (2005)

We have incorrectly cited Desroziers et al. (2006), and therefore replaced "Desroziers et al. (2006)" with "Desroziers et al. (2005)" in the third paragraph in Sect. 1, in the first sentence in subsection 2.1, and in References.

■ Reviewer 2

The authors propose to use a novel extension for the use of ensemble Kalman filters (EnKF) in a pre-operational ocean reanalysis product. The adaptive observation error inflation, previously introduced for satellite data assimilation, reduces assimilation increments by automatically inflating observational errors. The results show this automatic inflation as improvement compared to static observational errors. These results hold especially at Ocean frontal zones, where a large vertical diffusion can be observed with a static covariance. In general, this idea is relevant to improve data assimilation/reanalyses with ensemble Kalman filters, and the manuscript is well-written. Nevertheless, the manuscript needs a revision in its current form, at least with more and longer discussions, especially in relation to the number of figures. Also, the manuscript is not totally self-contained.

We thank the reviewer for constructive comments. We have replied to your comments in the following.

1) Whereas the ensemble Kalman filter, its assumptions, and its equations, are well-known, adaptive observation error inflation is quite unknown in the literature. Although the authors state and shortly explain the relevant equations, the explanations for this technique are too short. Its assumption and when we would expect that it works well remains totally unknown. Based on Desroziers et al., 2005 (often cited as 2005 and not 2006, which might confuse an informed reader), the relationship within the innovation statistics assumes Gaussian background and observational errors as in the ensemble Kalman filter, but what happens if these assumptions are violated?

We have incorrectly represented Desroziers et al. (2005) as Desroziers et al. (2006), and therefore replaced "Desroziers et al. (2006)" with "Desroziers et al. (2005)". As clear from Eq. (2), no correlation between forecast and observation errors is assumed in the formulation of the innovation statistics, and no assumption that the forecast and observation errors follow the Gaussian distribution is applied.

As indicated by the reviewer, the forecast and observation errors are assumed to follow the Gaussian distribution in the EnKF, but the effects of the forecast and observation errors not following the Gaussian distribution on the EnKF are beyond the scope of this study.

In addition, a crucial assumption is the correct representation of the background error

covariance with the ensemble; only then, Equation (1) represents a correct observational error covariance. The heavy use of relaxation to prior perturbations (RTPP) shows difficulties with the ensemble spread and I wonder if the ensemble spread is correctly tuned, especially in the Ocean frontal zones. The use of the maximum between estimated covariance and prescribed covariance lessens possible problems with these assumptions, but nevertheless, they should be named and discussed in the manuscript.

The equation from Desroziers et al. is only valid in expectation of the errors. For me, it remains unclear if and how this expectation is built in the data assimilation system. If no expectation is used, then its consequences and its connection to quality control and robust assimilation should be discussed, e.g., what happens in different innovation magnitude regimes (smaller or larger than the expected innovation magnitude)? In total, the method part of the adaptive observational error inflation needs to be revised.

As described in the first paragraph in subsection 2.2 in the original and revised manuscripts, we apply the perturbed atmospheric and lateral boundary conditions to the EnKF-based ocean data assimilation system to avoid filter divergence, following Ohishi et al. (in review) (See subsection 2.2 in Ohishi et al. in review). Ohishi et al. (in review) demonstrated that the combination of the IAU and RTPP with 80–90 % relaxation results in the best dynamical balance and accuracy, and therefore the RTPP parameter has been tuned in the experiments with fixed observation errors. However, because of the limitation of the computational resources, the tuning of the perturbed boundary conditions and RTPP parameter in the experiments with the AOEI is beyond the purpose of this study, as described in the last paragraph in Sect. 4.

We thank the reviewer for indicating statistical expectation and accuracy for the forecast ensemble spreads. We have added the statistical expectation to the LHS in Eq. (1). As indicated by the reviewer, the AOEI assumes that the forecast ensemble spreads are correct and the residual in Eq. (1) is caused by underestimation of the observation errors, and that $(d_b^o)^2$ is assumed to be equivalent to $\langle (d_b^o)^2 \rangle$ in Eq. (2). To mitigate underestimation of the estimated observation errors, the larger observation errors between the estimated and prescribed errors are chosen as shown in Eq. (3). We have added the assumptions used in the AOEI to the end of subsection 2.1.

2) The results show an improvement with adaptive observation error inflation compared to a static observational error assumption. The static observational errors results into too large assimilation increments and, thus, to a strong vertical diffusion at the Ocean frontal zones. As the static observational error covariances are important for increments, its

magnitudes are very important. Although the numbers are stated, their sources remains unknown. Because of the missing sources, the reader is unable to know if the prescribed uncertainties come only from the uncertainties of the observational products or if they also include other uncertainty sources like the observation operator or the representation error. The results indicate a larger representation error at the Ocean frontal zones than included in the observational error. A usual approach would be thus to generally inflate the observational errors or to withheld observations in these zones. Consequently, I would wish for a comparison experiment with an inflated observational error (e.g., 2 times the stated observational error) to see if a proper tuning of the errors would lead to better scores and how this might help in the case of the frontal zones. The authors have stated that they have only a limited computational budget, and a proper tuning of the observational errors and/or a comparison experiment might be too expensive. It might be therefore also enough to explain more in detail the advantages and disadvantages of adaptive observation error inflation compared to a tuned observational error, which can be again related to the discussion in point 1 of this review. Although the results seem to be good, the reader could be generally tempted to believe that the results are only caused by a non-tuned assimilation system.

We have compared the accuracy of the experiment with larger temperature observation error of 1.5 °C (denoted as 1.5Terr run hereafter) conducted in Ohishi et al. (in review), the CTL run with 1.0°C observation errors, and the AOEI run. For example, the RMSDs of the CTL, AOEI, and 1.5Terr runs relative to the drifter buoys are 0.260, 0.257, and 0.258 m s$^{-1}$ for surface zonal velocity, and 0.250, 0.248, and 0.249 m s$^{-1}$, respectively, and therefore the accuracy of the AOEI run is the best. We have added the description at the end of subsection 3.4 in the revised manuscript.

The following is my opinion for EnKF-based ocean data assimilation systems. Even if the perturbed boundary conditions are applied, the ensemble spread is small and assimilation impacts tend to be small in the subtropical region. To increase the assimilation impacts in the subtropical region, one might think that setting the smaller observation errors are better. However, small observation errors result in the degradation mechanisms in the frontal regions as seen in this study. The AOEI plays a role in suppressing the degradation mechanism if the small observation errors are set.

3) In general, the results part would profit a lot on concentrating on the most important parts of the study. Although well-written, the amount of figures compared to the discussion makes it difficult to follow the red line in the results part. Sometimes, similar

information is shown twice (e.g., Figure 6-8) and could be condensed into a single figure. Caused by the difficulties to follow the red line and a rather loose summary section, the main message of the manuscript remains also slightly unclear for me. On the one hand, this study tries to show how the static observational error induces problems with the vertical diffusion. On the other hand, it promotes of how adaptive observational error inflation can help. As discussed in section 2 of this review, the sensitivity experiments might be not enough to promote adaptive error inflation and to cancel out difficulties with the static observational error.

I like how the authors explain their evaluation in detail within the results part, but in its current form, it distracts from the main results and is too long. I would recommend to give here only concise explanations of the evaluation and to move specific equations and details into the appendix.

All of the descriptions, figures, and equations included in the manuscript are essential to reveal how the AOEI improves low-salinity structure. The degradation mechanism in the CTL run is quantitatively investigated in subsection 3.1, how frequency and where the AOEI is applied is shown in subsection 3.2, and the improvement mechanism by the AOEI is quantitatively investigated in subsection 3.3. The detail of the degradation mechanism in the CTL run and improvement mechanism by the AOEI would be useful for readers when they establish an EnKF-based ocean data assimilation system and face the similar problems. To clarify the story in Sect. 3, we have added the descriptions between Sect. 3 and subsection 3.1, and we have maintained the contents in the manuscript.

Smaller comments:

As an advice, the chosen colormaps might be generally misleading and inaccessible for colour-blind persons. In addition, the same colours are used for different meanings in subfigures (e.g. Figure 5) , which can be also very misleading for the reader.

As indicated by the reviewer, we have modified the color of the solid lines in Fig. 5a (11a) to distinguish between the colors of Fig. 5a (11a) and Fig. 5b, c (11b, c).

I would be interested into a comparison experiment without any data assimilation, except for example SST and SSH nudging as done for the spin-up phase. Currently, it remains

unclear for me if the noisier pattern in the SST fields compared to observations are caused by the data assimilation or if this is a "natural feature" of the model. This could be even shortly stated in the results part and then simply shown in a supplementary material or if this was discussed in the other manuscript, then the authors could simply point this fact to the other submission. In this sense also the naming of the experiments is a little bit confusing as the "control" run is usually an open-loop run without data assimilation whereas here it describes the baseline EnKF experiment, I would rename it into EnKF or STATIC.

>We have confirmed that the noisy SST and SSS signals and degradation of the low-salinity structure do not appear during the spin-up period. We have added the description at the end of the first paragraph in subsection 3.1.
>The CTL run is well used to compare between experiments with and without schemes even in data assimilation systems (e.g. Kotsuki et al. 2017; Zuo et al. 2019), and therefore we have maintained the name of the CTL run.

The authors frame the introduction as there are only two previous works on the EnKF for the Ocean. It might be correct that there are only two reanalysis products based on the EnKF but there is surely more work on the EnKF for the Ocean.

>As summarized in Ohishi et al. (in review), the EnKF is implemented with ocean data assimilation systems, but only two EnKF-based ocean reanalysis datasets exist to the best of our knowledge. To clarify that there are many ocean data assimilation systems with EnKF, we have added "(See table 1 of Ohishi et al. in review)" after "The EnKF has the advantage of being easy to implement for various models" in the first paragraph in Sect. 1.

In line 133, the authors state that they use covariance localisation. This term might be misleading, as they seem to use observational (covariance) localisation. I would rename it into R-matrix localisation as normally used in ensemble Kalman filter literature. In line 136, the use of incremental analysis updates (IAU) is indicated. The sentence links the use of IAU to ensemble inflation, which is not its normal use in ensemble Kalman filters. I would thus split the sentence with IAU and RTPP into two sentences. In addition, it is unclear how IAU is applied, if for example the increments are applied before and after the original time point or only after etc.

To clarify that covariance localization is applied in the observation space, we have added "in the observation space" after "Covariance localization" in the second paragraph in subsection 2.2.

As described in the original and revised manuscripts, Ohishi et al. demonstrated that the combination of the IAU and RTPP results in the best dynamical balance and accuracy, and therefore we have maintained that "the combination of the IAU (Bloom et al., 1996) and RTPP (Kotsuki et al., 2017; Zhang et al., 2004)". Here, Ohishi et al. (in review) have described the detail of how the IAU is applied, and therefore we have added "; Ohishi et al. in review" after "Bloom et al., 1996" in the second paragraph in subsection 2.2.

In line 143, "the" SSS nudging is named, what is "the" SSS nudging? Is it the same nudging as used for the spin-up phase? If yes, please state this explicitly.

We have used the same SSS nudging as the spin-up period in the CTL and AOEI runs, and added "as in the spin-up period" at the end of the last paragraph in subsection 2.2.

Other, smaller, issues could be resolved after a revision round.

Reference:
Ohishi S, Hihara T, Aiki H, Ishizaka J, Miyazawa, Y, Kachi M, and Miyoshi T.: An ensemble Kalman filter system with the Stony Brook Parallel Ocean Model v1.0, Geosci. Model Dev. Discuss. [preprint], https://doi.org/10.5194/gmd-2022-40, in review, 2022.
Zuo H, Balmaseda MA, Tietsche S, et al (2019) The ECMWF operational ensemble reanalysis–analysis system for ocean and sea ice: a description of the system and assessment. Ocean Sci 15:779–808. https://doi.org/10.5194/os-15-779-2019

---

## Author Response (AR2)

Before the reply to the reviewer, we must inform that we found a bug in a code to calculate the RMSDs relative to the KEO buoy and then corrected it (Fig. 12 in the revised manuscript). Specifically, the number in the denominator was smaller in the RMSD calculation, and the corrected results show larger RMSDs for all experiments than the previous results. However, the corrected results are qualitatively the same as the previous results. Consequently, this correction has few impacts on the conclusion in this paper. We apologize for the above.

**Referee #2**

Thank you for the response and the revision of the manuscript. The response and small changes have filled-in some missing pieces and clarified some points. However, I have still some remarks that might need clarification within the manuscript, especially related to the proper tuning of the experiments:

We thank the reviewer for reviewing in detail. We have added the verification results between CTL and 1.5Terr runs (reply to major comments #1 and #2) and the discussion about under-dispersive ensemble spreads relative to the RMSDs (major comment #3), and have reduced the amount of the descriptions and figures (major comment #4).

**1) In your response to comment #2 from the previous review #2, you have stated that the scores of the 1.5Terr experiment are better than for the CTL experiment, why do you use then the CTL experiment as baseline experiment? This result also indicate that the static observational error is not properly tuned (the representation error part might be missing). Consequently, the experiments are in some sense an unfair comparison between using a static observational error and AOEI. Furthermore, the aggregated score for the 1.5Terr experiment is very similar to the scores of the AOEI experiment. In the end, one could wonder if the AOEI experiment is really so much better than using a static observational error, if properly tuned. What is then the advantage of using AOEI?**

We have added the RMSDs of 1.5Terr and those relative to the Himawari-8 SSTs to Fig. 13 in the revised manuscript. The CTL run has significantly better accuracy for SSH and SST, whereas the accuracy of surface horizontal velocity and SSHA is degraded in the CTL run, but this is not significant. Since the Himawari-8 SSTs are not independent observations, we have also calculated the SST RMSDs relative to the independent KEO buoy. The results show the RMSDs of 0.52, 0.45, and 0.54 °C in the CTL, AOEI, and

1.5Terr runs, respectively, and therefore, the SST accuracy is better in the CTL run than in the 1.5Terr run. Namely, the 1.5Terr run does not have better accuracy for all variables compared with the CTL run. To highlight the impacts of the AOEI, we have chosen the experiment with temperature observation errors of 1.0°C rather than 1.5°C as the CTL run. We have added the related descriptions to the third paragraph in subsection 3.4.

Figure 13 in the revised manuscript shows that the AOEI run has the best accuracy for all variables except for SST. This paper demonstrates that the AOEI adaptively inflates the observation errors in the mid-latitude region, especially in the frontal region with the large representation errors, and suppresses the salinity degradation mechanisms in the CTL run. This results in better accuracy in the AOEI run than in the CTL and 1.5Terr runs with constant observation errors.

**2) Can you please insert the raw numbers for the 1.5Terr experiment in Fig. 17 and refer to this figure in line 367?**

We have added the results from the 1.5Terr run to Fig. 13 in the revised manuscript (Fig. 17 in the previous manuscript). Reference of Fig. 13 has been described at the end of the first sentence in the third paragraph in subsection 3.4 in the revised manuscript.

**3) As can be seen in Fig. 17, the RMSE is more than three times larger than the ensemble spread. Comparing Fig. 9 and Fig. 18, the variance error is the main driver for the RMSE. As now correctly stated, AOEI assumes a correctly tuned ensemble spread, which is here clearly not the case. I would like to see a discussion of this case (a paragraph or so) in the AOEI method part or section 3.4, as it is highly relevant for this study. I think you gave a starting point for this discussion in your response to comment #2.**

We thank the reviewer for indicating the large differences between RMSDs and ensemble spreads. As described in the third paragraph in subsection 2.2, Ohishi et al. (in review) performed the sensitivity experiments of covariance inflation and IAU methods and demonstrated that the combination of the IAU and RTPP with relaxing the analysis perturbations toward the forecast ensemble perturbations by 90% is the best for dynamical balance and accuracy. The large relaxation parameter plays a role in maintaining the ensemble spreads inflated by the perturbed atmospheric and lateral boundary conditions. Kurihara et al. (2016), for example, show that the RMSDs of the Himawari-8 SSTs relative to the buoys are about 0.5°C and larger in the higher latitude with a larger zenith

angle, and therefore, observation error variances might have substantial contributions to the RMSDs. Nevertheless, the ensemble spreads are much smaller than the RMSDs as indicated by the reviewer, likely being under-dispersive in this system.

We are now constructing analysis products using an eddy-resolving system with higher horizontal resolution of 0.1°, and the verification results show that the temporally averaged RMSDs of the surface horizontal velocity roughly correspond to the ensemble spread in the mid-latitude region, especially around the Kuroshio Extension region, whereas they do not in the subtropical region (figure not shown). Therefore, methods to inflate the ensemble spread more, especially in the subtropical region, are necessary but this will be a future topic. We have added the above discussion to the final paragraph in subsection 3.4.

**4) The added paragraphs at the beginning of the result sections have made them more readable. Nevertheless, I would highly recommend streamlining the discussion of the results. By its vast amount of information, the results are still difficult to follow. For example, by concentrating on the important parts of the Figures, Fig. 3 could be merged with Fig. 10. There is even a Figure (Fig. 18) that is not used at all.**

We have merged Figs. 3, 4, and 10 in the previous manuscript, which shows the zonal and meridional sections of temperature and salinity, into Fig. 3 in the revised manuscript, moved the results of the salinity budget analysis to Appendix, and removed relatively not crucial figure (Fig. 18 in the previous manuscript), equations [Eqs. (11), (13), and (14) in the previous manuscript], and related descriptions.

Minor points:

**1) Line 135: Please add a line break between the description of the assimilated observations and the description of the used parameters for the LETKF as the paragraph can be otherwise confusing.**

We have separated the descriptions of assimilated observation and parameters in LETKF, respectively, into the second and third paragraphs in subsection 2.2.

**2) Line 139: The source/citation for the observational errors is still missing, as indicated in the previous review.**

We have added the reference of Miyazawa et al. (2012). We note that the larger salinity observation errors are prescribed because the satellite observations appear to have large measurement errors.

**3) Line 384: Two tenses are used within that paragraph. I see the point in switching to past here for the summary within the summary. Nevertheless, I would stick to the present tense as the study still demonstrates the positive impact of AOEI.**

We have corrected from the past to present tense in the last sentence in the second paragraph of the revised manuscript.

**4) The chosen colormaps for continuous data can be still misleading and is difficult to see for colour-blind persons.**

We have modified the shading color of Figs. 1–3, 5–8, and 11 using scientific color maps based on Fabio Crameri (https://docs.generic-mapping-tools.org/6.2/cookbook/cpts.html).

**5) Fig. 13 (c-e): different blue tone in legend than in figures.**

We have modified the blue tone to be consistent with the lines in Fig. 9 in the revised manuscript (Fig. 13 in the previous manuscript).

**6) Fig. 14: Where is the heat flux within the figures? If it is covered by other lines (e.g., constant 0), please indicate this in the caption.**

We have added the description of "We note that the shortwave penetration gradient term is almost zero and is behind the diffusion gradient term in (a)–(c)." at the end of the caption of Fig. 10 in the revised manuscript (Fig. 14 in the previous manuscript).

**7) Fig. 17: The figures can be misleading. By plotting the ensemble spread together with the errors and using another axis, the reader could believe that the ensemble spread is similar to the errors, which is not the case.**

If the same ranges are used for the RMSDs and ensemble spreads, the differences between experiments are hard to be seen. We have added the description of "However,

the ensemble spreads are much smaller than the RMSDs for all variables (Fig. 13)." and the notes, respectively, in the last paragraph in subsection 3.4 and at the end of the caption of Fig. 13 in the revised manuscript (Fig. 17 in the previous manuscript).

**8) Fig. 18: Please use the same limits in the colormaps for the RMSE and spread; the figures are otherwise not comparable.**

Following major comment #4 from the reviewer, we have removed Fig. 18 in the previous manuscript.

---

## Author Response (AR3)

**Referee #1:**

I thank the Authors for all the corrections they have made in the new version of their manuscript, which improved it quite significantly. In my last comment, I would like to emphasise again that the impact of position errors should not be neglected in data assimilation results. This is an important issue to further improve data assimilation systems, especially as we move towards higher and higher resolution systems.

We thank the reviewer for reviewing again the manuscript. We have added the discussion of the importance of position errors for developing EnKF-based ocean data assimilation systems to the last paragraph in Section 4.

**Referee #2:**

Thank you for the additional revision of the manuscript. Now, your changes resolved my previously raised issues. Thank you also for adding the 1.5Terr experiment and changing the colormaps of the figures, they are now much more pleasant to look. I have still some smaller remarks, but beside this, the manuscript looks good:

We thank the reviewer for carefully reviewing the manuscript again. We have modified it following the reviewer's comments.

**1) Line 128: "assimilate the following observations" can be ambiguous, what is following referring to?. I supposed following the spin-up period, but remains unclear.**

We have modified the first sentence in the second paragraph in subsection 2.2.

**2) Line 138: why do you use for the localisation length-scale "LS" instead of the commonly used calligraphic l? Additionally, please use for the units in this line the non-mathematical font as used in Line 139.**

In the third paragraph in subsection 2.2, we have used $L$ for the localization scale instead of $LS$ following Houtekamer and Zhang (2016) and have modified the fonts of the units.

**3) Line 140f: please switch the position of salinity and SSH, as you speak afterwards about the salinity. This would improve the flow of reading.**

Following the reviewer's comment, we have modified it in the third paragraph in subsection 2.2.

**4) Line 170: I don't understand why the system conserves the temperature and salinity budget. With your increment from the EnKF you add/subtract temperature and salinity, so you don't converse the budget?**

Because of the effects of the covariance, temporally averaged nonlinear terms (ex. advection term) cannot be accurately calculated if temporally averaged single variables consisting of them (ex. temperature, salinity, and velocities) are used. This system accumulates each term in the temperature and salinity budget equations at each model timestep and each grid and then outputs the daily-mean values to close the budget. To

specify this, we have modified the last sentence in subsection 2.3.

**5) Line 240: "highest ... around" there is a preposition missing.**

We have inserted "at" between "the highest" and "around "35%–40%" in the second sentence in the first paragraph in subsection 3.2.

**6) Figure 4: Posit(i)ve, the i is missing in the legend.**

We have modified the legend in Fig. 4.

**7) Figure 5: There are white areas in (a)-(c), which indicate either missing values or ratios below your minimum colormap value. Please, either adapt the limit by extending the colormap or mention the white areas in the caption.**

We have added the explanation of white areas to the caption of Fig. 5.

**8) Figure 5 & 6: you use the same colormap for panel d and e, although they present different quantities. Could you please choose another map for one of them, otherwise the reader might be confused.**

We have modified the color pallets in Figs. 5e and 6e.

**9) Figure 7 (b): The unit in the bias is missing. If the bias of the SSS is dimensionless, then please use "Bias (1)". Otherwise, the reader could be misled and think that you simply made a copy and paste error from panel (a), since you use exactly the same colormap and limits.**

To clarify that Fig. 7a and 7b shows the different variables, we have modified the color pallet in Fig. 7b.

**10) In your equations, you write out "increment". For an informed reader, it is clear that you mean the increment of this variable, but it remains nevertheless ambiguous, if it means only the increment of the specific variables or the increment of all variables. Consequently, I would suggest using the common delta formulation, e.g. $\Delta T$ for the increment in the temperature.**

To clarify which variable is used for the increment terms, we have replaced $(increment)$ with $(T\ increment)$ and $(S\ increment)$ in the temperature and salinity budget equations [Eqs. (4), (5)], respectively, as well as the related equations [Eqs. (13), (14), and (B1)] and have modified the related descriptions.

**11) You quite often refer to not-shown figures. Although the figures might not give too much new information, you partially base your arguments on them. Thus, I would prefer to see them in the Appendix.**

By providing the four sentences including "(figure not shown)" with additional sufficient explanation and information, we have removed "(figure not shown)".